# Local H$_2$ release remodels senescence microenvironment for improved repair of injured bone

Shengqiang Chen[1,2,7], Yuanman Yu [3,7], Songqing Xie[1,7], Danna Liang[4], Wei Shi[1], Sizhen Chen[1], Guanglin Li[1], Wei Tang [1] ✉, Changsheng Liu [3] ✉ & Qianjun He [2,5,6] ✉

The senescence microenvironment, which causes persistent inflammation and loss of intrinsic regenerative abilities, is a main obstacle to effective tissue repair in elderly individuals. In this work, we find that local H$_2$ supply can remodel the senescence microenvironment by anti-inflammation and anti-senescence effects in various senescent cells from skeletally mature bone. We construct a H$_2$-releasing scaffold which can release high-dosage H$_2$ (911 mL/g, up to 1 week) by electrospraying polyhydroxyalkanoate-encapsulated CaSi$_2$ nanoparticles onto mesoporous bioactive glass. We demonstrate efficient remodeling of the microenvironment and enhanced repair of critical-size bone defects in an aged mouse model. Mechanistically, we reveal that local H$_2$ release alters the microenvironment from pro-inflammation to anti-inflammation by senescent macrophages repolarization and secretome change. We also show that H$_2$ alleviates the progression of aging/injury-superposed senescence, facilitates the recruitment of endogenous cells and the preservation of their regeneration capability, thereby creating a pro-regenerative microenvironment able to support bone defect regeneration.

Increasing cellular senescence is one of major reasons of causing aging and many diseases[1–3]. Senescence is characterized by a persistent cell-cycle arrest and a distinctive pro-inflammatory secretory phenotype of various cells, which create a senescence microenvironment (SME) to drive tissue dysfunction towards deterioration within a negative feedback loop and thus to decline the self-repair ability of tissues[4,5]. Typically, SME causes a bone loss during aging and a huge challenge in the repair of bone fracture/defect of elders[2,6]. With the increase of aging population, to improve or even remodel SME for addressing the issue of injured aging bone repair becomes increasingly urgent, but remains greatly challenging.

Generally, two types of anti-senescence strategies have been developed to manage the senescence-related diseases such as aging bone diseases and cancers, including the lysis of senescent cells using senolytic drugs (senolytics) and the morphing of the senescence-associated secretory phenotype (SASP) using senomorphic drugs (senomorphics)[1,4,7–9]. Unfortunately, none of drugs can effectively regulate the whole SME, which is involved in varied types of senescent

[1]Key Laboratory of Human-Machine-Intelligence Synergic System, Research Center for Neural Engineering, Shenzhen Institute of Advanced Technology, Chinese Academy of Sciences, Shenzhen 518055 Guangdong, China. [2]Shanghai Key Laboratory of Hydrogen Science & Center of Hydrogen Science, School of Materials Science and Engineering, Shanghai Jiao Tong University, Shanghai 200240, China. [3]The State Key Laboratory of Bioreactor Engineering, East China University of Science and Technology, Shanghai 200237, China. [4]Guangdong Key Laboratory for Biomedical Measurements and Ultrasound Imaging, School of Biomedical Engineering, Health Science Center, Shenzhen University, Shenzhen 518060, China. [5]Medical Center on Aging, Ruijin Hospital, Shanghai Jiao Tong University School of Medicine, Shanghai 200025, China. [6]Shenzhen Research Institute, Shanghai Jiao Tong University, Shenzhen 518057, China. [7]These authors contributed equally: Shengqiang Chen, Yuanman Yu, Songqing Xie. ✉e-mail: wei.tang1@siat.ac.cn; liucs@ecust.edu.cn; nanoflower@126.com

cells with different SASPs, due to the lack of both anti-senescence selectivity and universality of existing drugs[4,9–11], leading to obvious toxic side effects and limited anti-senescence efficacies.

But strikingly, hydrogen molecule (H$_2$) has proven to be an emerging safe, valid and broad-spectrum anti-inflammatory agent owing to its anti-oxidation ability for selectively scavenging highly oxidative/toxic radicals such as hydroxyl radicals (·OH)[12–14]. Since senescence is closely associated with oxidative stress, H$_2$ was found to have an anti-senescence effect to many cells (bone marrow-derived mesenchymal stem cells (BMSCs), fibroblasts, and endothelial cells) and tissues (brain, periodontium, aorta, and retina) without obvious toxic side effects, demonstrating excellent selectivity and universality of anti-senescence[15–22]. However, aging/injury-superposed senescence is much more serious than individual aging- or injury-induced one, and the effect of H$_2$ on aging/injury-concurrent senescence from aging bone injury is unknown so far. Moreover, the previously reported H$_2$ administration routes for anti-senescence mainly including oral uptake of hydrogen-rich water and inhalation of hydrogen gas cannot achieve a high H$_2$ concentration for a long time duration at the diseased site owing to low solubility and high dispersion of H$_2$, causing limited anti-senescence efficacies.

In this work, we revealed the necessity of sustained H$_2$ supply for remodeling the SME in aging bone firstly, and then designed a kind of polyhydroxyalkanoate (PHA)-encapsulated CaSi$_2$ nanoparticles (CSN)-loading mesoporous bioactive glass (MBG) scaffolds (CSN@PHA-MBG) for local and sustained (up to 1 week) release of high-dosage H$_2$ (911 mL/g CSN), realizing efficient remodeling of SME and enhanced repair of injured aging bone. The hydrogen storage capacity of CSN was $4.6 \times 10^4$ folds higher than that of water, and sustained H$_2$ release capability (about 1 week) of CSN@PHA was far higher than that of hydrogen-rich water (about 30 min). A mechanism of SME remodeling for injured aging bone repair was discovered that sustained H$_2$ treatment universally attenuated oxidative stress in the SME and efficiently remodeled the SASPs of various senescent cells by an anti-inflammation pathway, inducing macrophage repolarization to an anti-inflammatory phenotype, BMSCs recruitment, angiogenesis and osteogenesis in support of injured aging bone repair. The proposed local H$_2$-releasing strategy provides a promising anti-senescence approach for bone regeneration.

## Results

### Sustained H$_2$ supply enables anti-inflammation-mediated anti-senescence

Senescent cells typically exhibit DNA damage, cell cycle blockade, metabolic dysfunction of mitochondria and lysosome, and secretome change, which are characterized with the up-regulations of senescence-associated intracellular reactive oxygen species (ROS), p21/p16 proteins (cell cycle inhibitory proteins), β-galactosidase (SA-β-gal, a biomarker of senescence in lysosomes), colocalization of γ-H2A.X (a biomarker of DNA double-strand breaks) with telomere-associated foci (TAFs), and SASP components, as well as with the down-regulations of heme oxygenase-1 (HO-1, a cytoprotective enzyme) and Ki67 (a biomarker of proliferation)[23–25]. In addition, ROS is highly correlated with senescence owing to its adverse effects in driving a vicious cycle[26,27], and can be effectively scavenged by H$_2$[12,13]. Therefore, these representative indicants were detected to investigate the effect of sustained H$_2$ supply on the senescence of various cells from aging bone. BMSCs, macrophages and osteocytes were isolated from the bone of 24-month-old aging mice (equal to 70-year-old people) for identification (Fig. 1a, Supplementary Figs. 1 and 2), and continuously incubated in a 60% hydrogen incubator for 7 days, which was used to simulate a sustained H$_2$ supply microenvironment in vitro.

Compared to the cells harvested from young bone, all the investigated types of cells harvested from aging bone exhibited obvious senescence characteristics, as indicated by excessive expression of intracellular ROS, p16, p21, and SA-β-gal, as well as lower expression of HO-1 and Ki67 (Fig. 1b and Supplementary Figs. 3–5). Moreover, the γ-H2A.X foci were found to colocalize with TAFs (Fig. 1b). The continuous 7-day treatment with H$_2$ significantly down-regulated the expressions of all these senescence biomarkers and further promoted the HO-1 activity and Ki67 expression in all the investigated cells (Supplementary Fig. 6), suggesting a potent universal anti-senescence effect of H$_2$. On the other hand, senescent cells secret large amounts of SASP components including inflammatory molecules, chemokines and degradative enzymes, creating a SME to make neighboring normal cells senescent[28,29]. Generally, H$_2$ lowered the levels of IL-6 and IL-1β, two of representative SASP components in all the investigated types of cells (Fig. 1b). In particular, H$_2$ significantly down-regulated the IL-6 expression at both day 3 and day 7 in all the investigated types of cells. Similarly, H$_2$ remarkably lowered the IL-1β expression in BMSCs and macrophages, especially after continuous 7-day treatment. IL-6 and IL-1β are two potent pro-inflammatory cytokines, which can establish a pro-inflammatory microenvironment in support of persistent inflammation and senescence extension[3,27,30]. To determine the role of H$_2$ in regulating the inflammation of senescent cells, the effect of H$_2$ treatment on the M2 polarization of senescent macrophages was evaluated. As shown in Supplementary Fig. 7, H$_2$ polarized the macrophages from a M1 phenotype (CD86 and *iNOS*) to a M2 phenotype (CD206, *Arg1* and *Il-10*) during 7 days of treatment. From Supplementary Fig. 8, the ratio of TNF-α level to TGF-β1 level sharply declined to 0.21 and 0.13 from 0.47 and 1.05 after H$_2$ treatment for 3 and 7 days, respectively, indicating a H$_2$-induced great shift from M1 to M2 subtype of macrophages. Moreover, we further investigated the influence of H$_2$ on the SA-β-gal expression and cell viability of senescent osteocytes from 24-month-old mice, and found that H$_2$ had both senomorphic and senolytic effects and played a primary role of senomorphic agent (Supplementary Fig. 9).

Since senescence will drive the loss in the regenerative function of stem cells, we next checked whether H$_2$ can rescue the regenerative potential of senescent stem cells by preventing or inhibiting their senescence. Indeed, BMSCs acquired an increasing senescence phenotype with the increase of passage number, as determined by SA-β-gal staining results, and their proliferation was arrested at later passages (Supplementary Fig. 10). By contrast, H$_2$ treatment markedly prevented the progression of BMSCs senescence, as revealed by increasing reduction of SA-β-gal-positive cells with time (Supplementary Fig. 11) and significant lower expression of senescence-related genes (*p16*, *p21*, and *p53*, Supplementary Fig. 12) in senescent BMSCs, meaning that sustained supply of H$_2$ held a potential of senescence inhibition. Furthermore, H$_2$ significantly suppressed the effects of senescent cells on osteoclastogenesis, as suggested by that conditioned medium from senescent cells without H$_2$ treatment induced the increases in the number of tartrate-resistant acid phosphatase (TRAP)-positive cells and in the enzyme activity of TRAP, but H$_2$ treatment considerably suppressed such increases (Supplementary Fig. 13). On the other hand, the senescence-impaired osteogenic differentiation potential of BMSCs was preserved by H$_2$, as indicated by enhanced expression of osteogenesis-related genes (*Runx2*, *Osterix*, *Alp*, *Opn*, *Col.I* and *Ocn*, Supplementary Fig. 14a, b) and higher degree of mineralization (alizarin red staining, Supplementary Fig. 14c). Taken together, sustained supply of H$_2$ can inhibit the osteoclastogenesis and also preserve the regenerative capability of BMSCs by preventing their senescence, which was expected to favor injured aging bone repair.

### Construction and characterization of the sustained H$_2$-releasing scaffold

The above discovery about the anti-inflammatory, SME-remodeling and regenerability-preserving potentials of sustained H$_2$ supply inspired us to develop a kind of scaffolds in competence of high H$_2$ loading and sustained H$_2$ release for injured aging bone repair. To

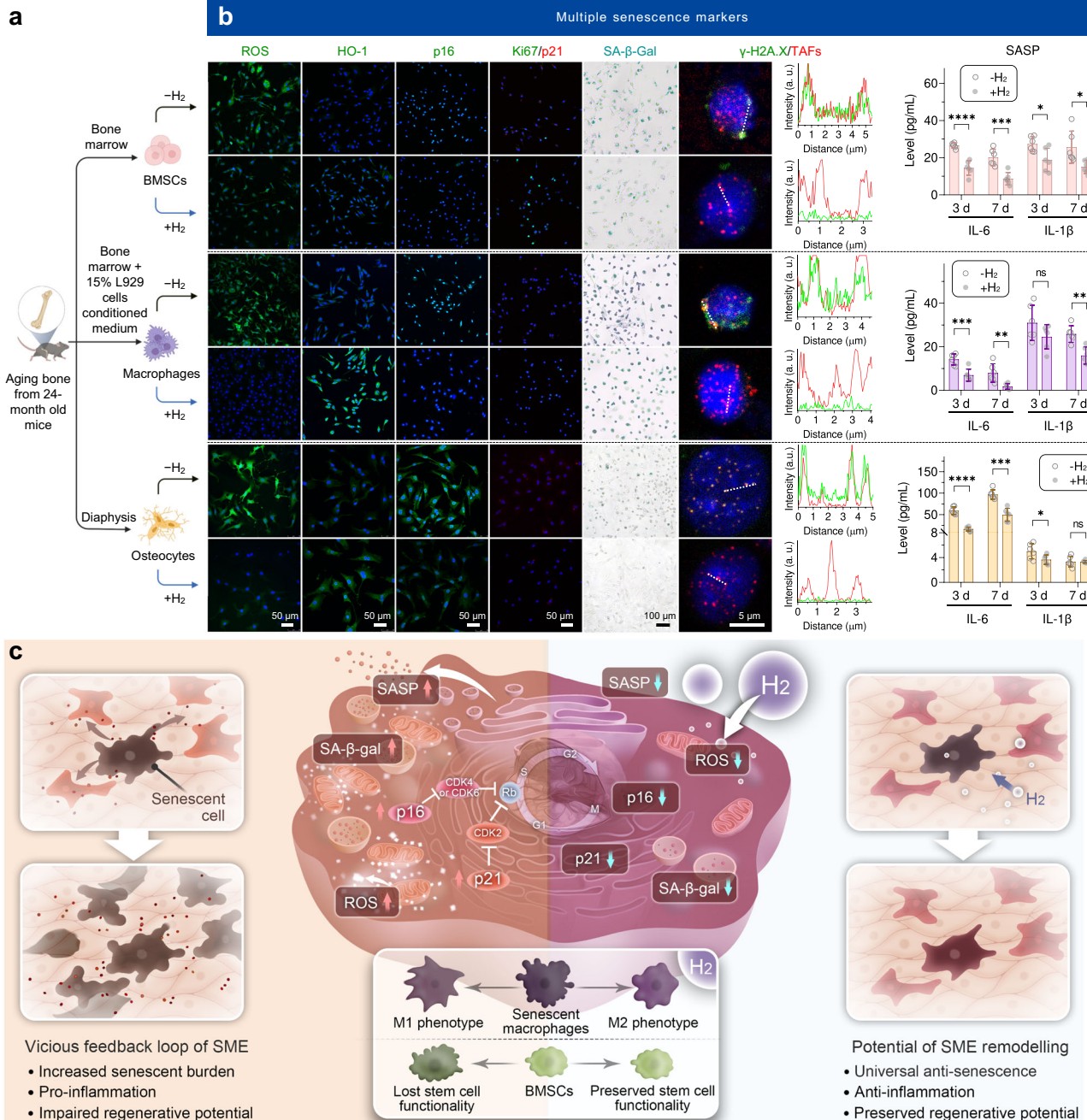

**Fig. 1 | In vitro anti-inflammation and anti-senescence behaviors of sustainable H₂ treatment of various senescent cells from aging bone.** Schematic illustration of testing the effects of sustained H₂ treatment on BMSCs, macrophages and osteocytes collected from aging bone (**a**), the measurement of representative senescence biomarkers (ROS, HO-1, p16, p21, Ki67, γ-H2A.X, TAFs, SA-β-gal, IL-6, and IL-1β) in BMSCs, macrophages and osteocytes continuously incubated with or without H₂ for 7 days (*n* = 6 biologically independent samples for ELISA measurement) (**b**), illustration of cellular senescence features and the anti-senescence, anti-inflammation, and regeneration-preserving effects of H₂ for SME modulation (**c**).

Data are means ± SD. *P* values in figure b were calculated by the two-tailed unpaired Student's *t* test method (BMSCs: ****$p < 0.0001$ in day 3 for IL-6, ***$p = 0.0005$ in day 7 for IL-6, *$p = 0.0181$ in day 3 for IL-1β, *$p = 0.0193$ in day 7 for IL-1β; macrophages: ***$p = 0.0008$ in day 3 for IL-6, **$p = 0.0065$ in day 7 for IL-6, ns in day 3 for IL-1β, **$p = 0.0012$ in day 7 for IL-1β; osteocytes: ****$p < 0.0001$ in day 3 for IL-6, ***$p = 0.0001$ in day 7 for IL-6, *$p = 0.0454$ in day 3 for IL-1β, ns in day 7 for IL-1β; ns means no significant difference ($p > 0.05$)). **a** is created with BioRender.com. Source data are provided as a Source Data file.

address this, CaSi₂ was here recognized as a potential hydrogen prodrug with a high H₂ productivity (104.2 mg/g, or 1.28 L/g) and a high biocompatibility based on easy metabolism and osteoinductivity of its hydrolysis products Ca²⁺ and SiO₃²⁻. But CaSi₂ bulk and micrometer powder are hardly decomposed in the physiological conditions. Therefore, we here hypothesized that nanosizing could enhance the hydrolytic activity of CaSi₂ for hydrogen release.

A microwave-assisted chemical stripping method was employed to synthesize CaSi₂ nanoparticles (CSN). SEM images (Fig. 2a, b and Supplementary Fig. 15) indicated that the size of synthesized CSN was about 500 nm. EDS elemental mapping (Fig. 2b) and XRD data (Fig. 2c) clearly confirmed that chemical stripping had not changed the chemical composition and crystal structure of CaSi₂. Moreover, CSN did not cause observable cytotoxicity to BMSCs within a wide CSN

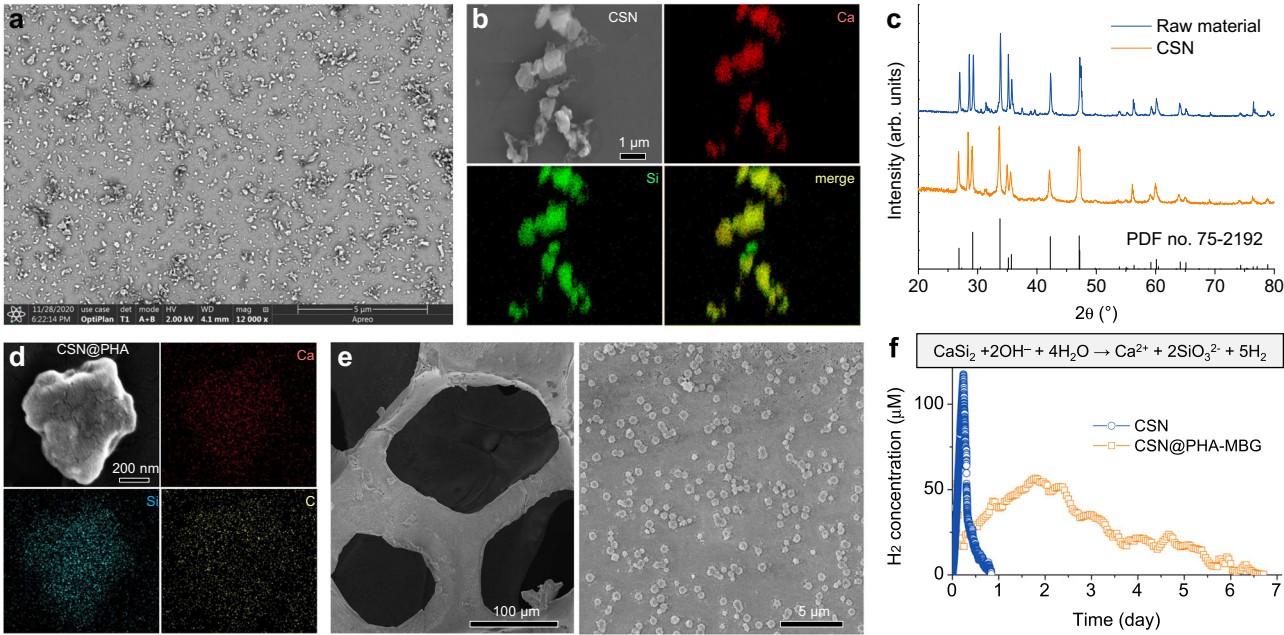

**Fig. 2 | The structures and hydrogen release behaviors of CSN and CSN@PHA-MBG scaffold.** Representative SEM (**a**) and elementary mapping (**b**) images of CSN, XRD patterns of CSN and CaSi$_2$ raw material (**c**), representative SEM and elementary mapping images of CSN@PHA (**d**), SEM images of the CSN@PHA-MBG scaffold with different magnifiication times (**e**), and the H$_2$ release profiles of CSN and CSN@PHA-MBG with a CSN content of 1 mg mL$^{-1}$ in the PBS (**f**). The insert shows the reaction of CSN in the aqueous solution with a high H$_2$ yield. All the experiments were repeated three times independently with similar results. Source data are provided as a Source Data file.

concentration range, exhibiting high cytocompatibility (Supplementary Fig. 16). Excitingly, CSN can indeed be hydrolyzed to release a high amount of H$_2$ in the PBS with pH = 7.4 as expected, but will be degraded completely within one day (Fig. 2f and Supplementary Fig. 17). However, early-stage inflammation in bone injury will last several days, and sustained H$_2$ supply for several days was also confirmed above to be favorable for aging bone injury repair (Fig. 1). Therefore, to achieve a sustainable H$_2$ release, CSN was wrapped by a layer of poly-hydroxyalkanoate (PHA, a natural biopolymer) to reduce the rate of CSN hydrolysis/degradation, and then deposited on the porous surface of MBG substrate by an electrospray method. Besides high biocompatibility and good biodegradability, the main rationale for choosing PHA lied in its hydrophobicity and thermoplasticity in support of impeding CSN hydrolysis for sustained H$_2$ release and enhancing the electrospray practicability of CSN encapsulated polymer nanoparticles, respectively. In addition, compared to poly (lactide-*co*-glycolide) (PLGA), PHA has lower biodegradability in favor of sustained CSN hydrolysis (H$_2$ release), and its degradation products exhibit higher tissue compatibility[31,32].

At an optimized parameter, CSN@PHA can be electrically sprayed into regular particles with uniform particle size (Supplementary Figs. 18 and 19). Elemental mapping patterns indicated that CSN could be embedded in PHA in a one-encapsulated-one manner (Fig. 2d). After electrospray, it can be observed that CSN@PHA was uniformly coated on the porous surface of MBG (Fig. 2e). Notably, the synthesized CSN@PHA-MBG exhibited a sustained H$_2$ release profile and its release time lasted up to 7 days in spite of the CSN encapsulating content (Fig. 2f, Supplementary Figs. 17 and 20). Such a prolonged H$_2$ release duration had not been realized by previously reported hydrogen donors, and will be able to cover the inflammation and angiogenesis periods of bone regeneration[33,34], and also be particularly beneficial for early-stage SME remodeling in aging bone injury as indicated above (Fig. 1). Furthermore, live/dead cell staining revealed that both PHA-MBG and CSN@PHA-MBG scaffolds with different CSN contents had high cytocompatibility, as indicated by no visible influence on the viability of BMSCs (Supplementary Fig. 21). More importantly, after

7 days of culture, the CSN@PHA-MBG scaffolds resulted in markedly lower SA-β-gal expression in BMSCs compared to the PHA-MBG scaffold without H$_2$ supply in a CSN content-dependent way, exhibiting a minimum effective CSN content (or hydrogen dose) of about 70 μg g$^{-1}$ (Supplementary Fig. 22). The CSN@PHA-MBG scaffold with a CSN content of 560 μg g$^{-1}$ was used for following in vivo therapeutic experiments. The developed H$_2$-releasing CSN@PHA-MBG enabled local and sustainable supply of high-dosage H$_2$, providing a possibility for promoting injured aging bone regeneration.

## Local sustained hydrogen release remodels the SME and polarizes macrophages towards an anti-inflammatory phenotype in the injured aging bone

In view of in vitro profound effects of sustained H$_2$ release in universal anti-inflammation/senescence, the in vivo therapeutic principle and performance of the CSN@PHA-MBG scaffold were further evaluated using an aged mouse (24-month-old) critical-size femoral plug defect model (1.2 mm-diameter round defect) (Fig. 3a). Since chronic inflammation in the SME hugely impedes aging tissue healing and the early-stage anti-inflammation in bone injury is vitally important to in-time and effective bone repair[5,33], we firstly investigated the in vivo hydrogen release behavior of the CSN@PHA-MBG scaffold and the effect of the CSN@PHA-MBG scaffold on local inflammation within 7 days after treatment. From Supplementary Fig. 23, it can be found that the in vivo hydrogen release time of the CSN@PHA-MBG scaffold was as long as 9 days, even longer than that in the PBS (pH = 7.4), which was possibly owing to the slower diffusion rate of hydrogen molecules in bone. Such a sustained hydrogen release behavior would favor local anti-senescence and anti-inflammation. From Fig. 3b, c, intensive green fluorescence of F4/80 (a biomarker of macrophages) appeared at the implant site at day 3 and 7, reflecting a typical phenomenon of inflammation in the initial stage of bone repair when macrophages were largely recruited. Although there were no significant differences in the extent of macrophages infiltration, the CSN@PHA-MBG scaffold exhibited a potent anti-inflammatory effect compared with control and PHA-MBG carrier, as indicated by lower iNOS expression (a

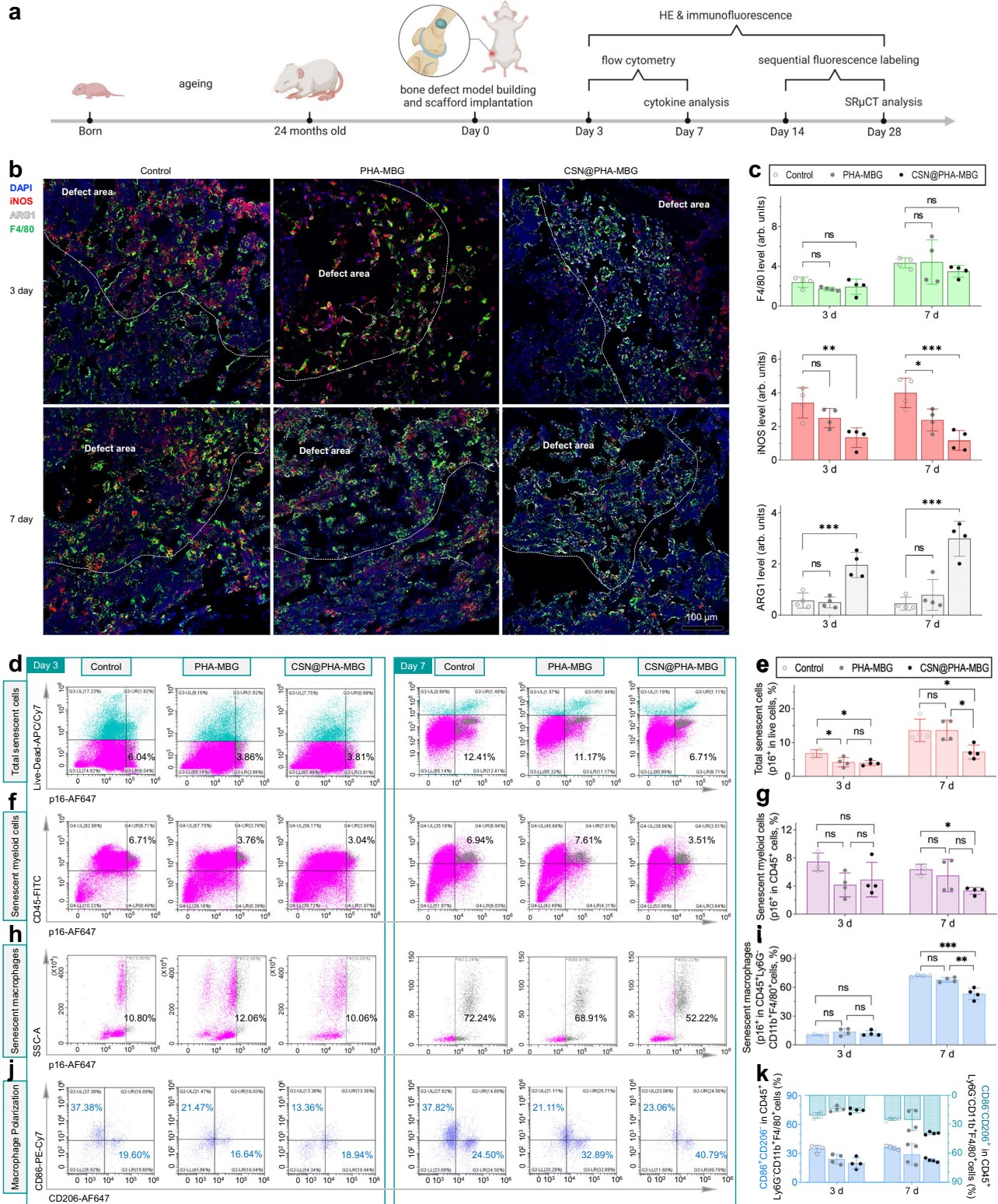

**Fig. 3 | The effect of the CSN@PHA-MBG scaffold on cellular senescence and macrophage polarization in the injured aging bone.** Schematic illustration of the in vivo therapeutic and measurement schedule (**a**), representative immunostaining images (**b**) and corresponding quantification analyses (**c**) of F4/80, iNOS and ARG1 levels at the implant site (*n* = 4 biologically independent samples), representative flow cytometry plots (**d**, **f**, **h**, **j**) and corresponding quantification analyses (**e**, **g**, **i**, **k**) (*n* = 4 biologically independent samples) of total senescent cells (**d**, **e**), senescent myeloid cells (**f**, **g**), senescent macrophages (**h**, **i**), and macrophage polarization

(**j**, **k**) (*n* = 4). Data are means ± SD. *P* values were calculated by the one-way ANOVA method with Tukey's post hoc test. **c** ***p* = 0.0063 in day 3 for iNOS, **p* = 0.0274, ****p* = 0.0009 in day 7 for iNOS; ****p* = 0.0009 in day 3 for ARG1, ****p* = 0.0003 in day 7 for ARG1. **e** **p* = 0.0156, **p* = 0.0251 in day 3; **p* = 0.0273, **p* = 0.0260 in day 7. **g** **p* = 0.0341 in day 7. **i** ****p* = 0.0002, ***p* = 0.0011 in day 7; ns means no significant difference (*p* > 0.05). **a** is created with BioRender.com. Source data are provided as a Source Data file.

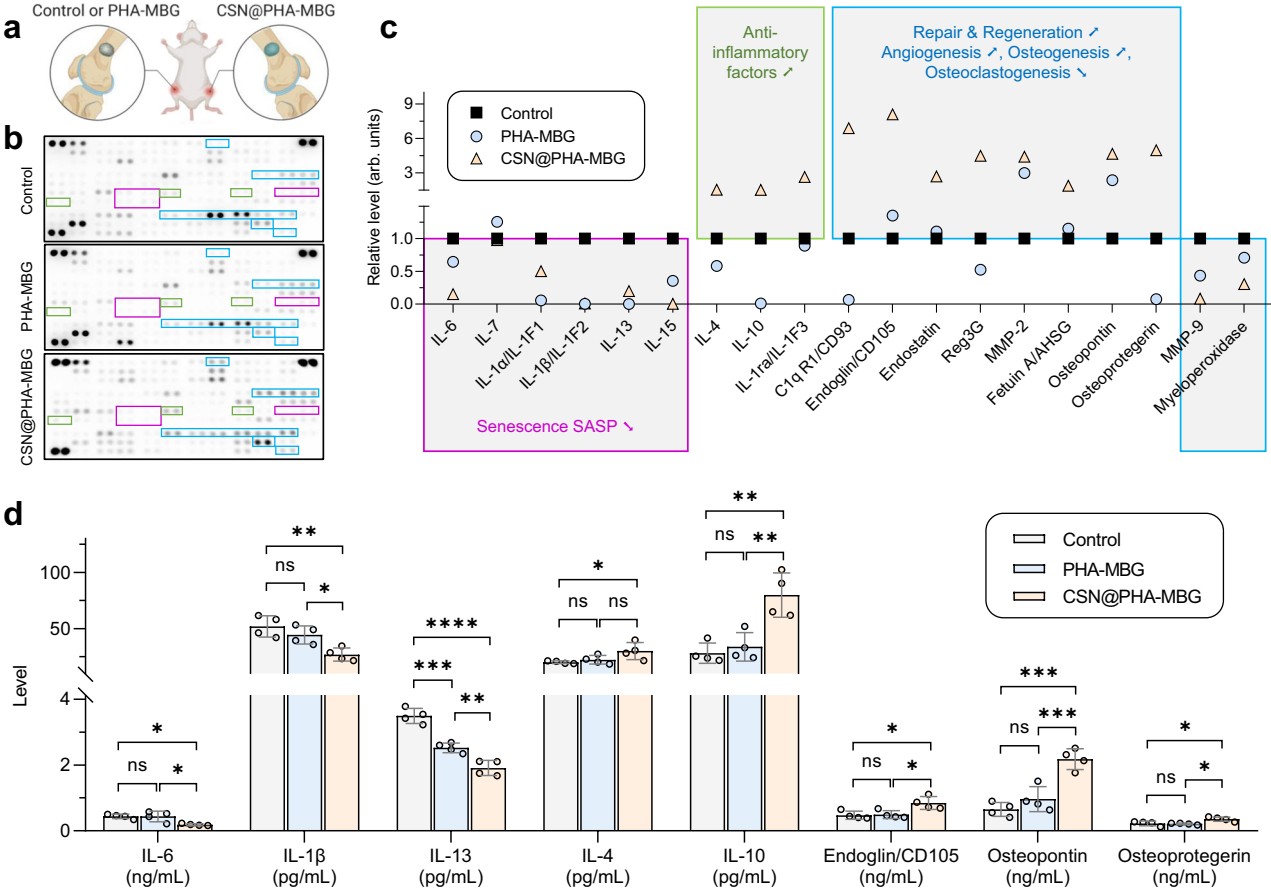

**Fig. 4 | SASP components analysis at the aging bone defect site after scaffold treatment for 7 days.** Schematic illustration of the experimental design for cytokine array analysis (**a**), representative mouse cytokine array images (**b**) and corresponding quantification (**c**) (*n* = 2 biologically independent samples), and cytokine levels measured by ELISA (**d**) (*n* = 4 biologically independent samples). Data are means ± SD. *P* values were calculated by the one-way ANOVA method with Tukey's

post hoc test. **d** \**p* = 0.0167, \**p* = 0.0168 for IL-6; \*\**p* = 0.0040, \**p* = 0.0302 for IL-1β; \*\*\*\**p* < 0.0001, \*\*\**p* = 0.0003, \*\**p* = 0.0059 for IL-13; \**p* = 0.0446 for IL-4; \*\**p* = 0.0018, \*\**p* = 0.0039 for IL-10; \**p* = 0.0155, \**p* = 0.0209 for CD105; \*\*\**p* = 0.0002, \*\*\**p* = 0.0009 for Osteopontin; \**p* = 0.0253, \**p* = 0.0149 for Osteoprotegerin; ns means no significant difference (*P* > 0.05). **a** is created with BioRender.com. Source data are provided as a Source Data file.

biomarker of M1 macrophage phenotype), higher ARG1 level (a biomarker of M2 macrophage phenotype) and less fibrosis (Supplementary Fig. 24), mainly owing to sustained hydrogen release.

Furthermore, flow cytometric analysis of cells from the aging bone defect site was conducted to evaluate the role of local sustained $H_2$ release in modulating senescence and inflammation in the SME (Fig. 3d−k and Supplementary Figs. 25−29). From Fig. 3d, e, p16-positive cells occupied a high proportion which increased with time (6.04% on day 3, and 12.41% on day 7), displaying a typical bystander effect that senescence gradually spread to surrounding cells[7,27]. Even so, CSN@PHA-MBG considerably and steadily alleviated the progression of aging/injury-superposed senescence, but PHA-MBG as a carrier always did not so significantly (Fig. 3d−g), suggesting that sustained $H_2$ release from CSN@PHA-MBG can indeed locally prevent cellular senescence persistently in the SME of aging bone defect in accordance with in vitro results (Fig. 1). The in vivo immuno-florescence in-situ hybridization (FISH) results indicated a similar phenomena that CSN@PHA-MBG caused less DNA damage and colocalization of γ-H2A.X with TAFs than other groups (Supplementary Fig. 30). Especially, in accordance with previous reports[35], macrophages in the SME exhibited a high p16-positive phenotype (Fig. 3h, i). Although the $H_2$-induced reduction in the proportion of p16-positive macrophages on day 3 seemed not so remarkable, sustained $H_2$ release up to 7 days played an important role in preventing macrophages senescence and mediating immune responses in a macrophage phenotype dependent manner. From Fig. 3j,k, the number of M1-phenotype macrophages in

the control group was much higher than that of M2-phenotype ones (approximate 2 folds on day 3), reflecting a highly pro-inflammatory response in the inflammation stage. CSN@PHA-MBG more remarkably and steadily reduced the ratio of macrophage numbers between M1 phenotype and M2 phenotype (0.71 on day 3, and 0.57 on day 7), compared with the control (1.91 on day 3, and 1.54 on day 7) and PHA-MBG (1.29 on day 3, and 0.64 on day 7), indicating that sustained $H_2$ release made a major contribution to anti-inflammatory polarization in accordance with in vitro results (Fig. 1). Especially on day 7, CSN@PHA-MBG induced a high percentage of M2-phenotype macrophages infiltration (about 40%, Fig. 3j, k), implying the effective transformation of microenvironment from pro-inflammation to anti-inflammation in favor of SME remodeling and aging bone defect repair.

The cytokines and chemokines derived from local cells play an important role in manipulating microenvironment and immune response. Due to the accumulation of senescent cells with aging, senescent cells can develop SASP and extend their deleterious paracrine secretion to surrounding cells, leading to chronic inflammation and impeded tissue repair. In view of these, we analyzed the expression of cytokines in the injured tissue collected at the initial inflammatory stage by cytokine array and ELISA (Fig. 4a). As demonstrated in Fig. 4b, c, significantly differentiated expression of cytokines between the control group and scaffold treatment groups can be classified mainly into three categories including down-regulation of SASP components (violet panels), up-regulation of anti-inflammatory factors (green panels), and improvement of repair/regeneration-related

factors (blue panels). It can be clearly visible that typical SASP components including IL-6, IL-7, IL-1α, IL-1β, IL-13 and IL-15 were remarkably depressed by the CSN@PHA-MBG scaffold, while representative anti-inflammatory factors including IL-4, IL-10 and IL-1ra/IL-1F3 were regulated up by the CSN@PHA-MBG scaffold. CSN@PHA-MBG generally exhibited a stronger and more positive regulation effect than PHA-MBG, implying that sustained hydrogen release made a major contribution compared with the carrier. Furthermore, from Fig. 4c, compared with PHA-MBG, CSN@PHA-MBG more significantly induced the up-regulation of typical angiogenesis (C1qR1/CD93, and Endoglin/CD105) and osteogenesis (Reg3G, MMP-2, Fetuin A, Osteopontin and Osteoprotegerin) factors, and also inhibited the expression of typical osteoclastogenic factors (MMP-9 and Myeloperoxidase). To give a more accurate quantification, prototypical components of SASP, anti-inflammatory cytokins and osteogenesis-related proteins from the aging bone defect site were analyzed using ELISA. In accordance with in vitro results (Fig. 1), sustained $H_2$ supply in the SME can significantly attenuate SASP involving declined levels of IL-6, IL-1β, and IL-13 (Fig. 4d), and serve as an anti-inflammatory modulator with increased IL-4 and IL-10 levels. Concurrently, significant higher expressions of Endoglin/CD105, osteopontin and anti-osteoclastogenic cytokine osteoprotegerin were observed in CSN@PHA-MBG compared to control and PHA-MBG, indicating that local sustained $H_2$ release from CSN@PHA-MBG can indeed regulate the secretome in the SME to favor bone regeneration. By combination of universal cellular senescence, inflammation and SASP analyses, it can be confirmed that the SME at the aging bone defect site had been remodeled by sustained hydrogen release from the CSN@PHA-MBG scaffold.

## The hydrogen-releasing scaffold creates a pro-regenerative microenvironment at the aging bone defect site based on the anti-senescence effect of $H_2$

Generally, bone regeneration is a coordinated process, which is involved in the distinct overlapping of several typical events including haemostasis and inflammation, neo-tissue formation (recruitment of endogenous cells) and tissue remodeling[33]. Moreover, there is a close link between cell senescence and tissue regeneration[28]. Since local sustained $H_2$ release from CSN@PHA-MBG can indeed remodel the SME and play an anti-inflammatory role in the SME in the initial stage of bone repair, we next sought to explore whether such effects would form a positive feed-forward loop in facilitating regeneration by preventing endogenous stem cell senescence and activating their regenerative functions. To check this, we performed the co-staining of injured bone tissues with p16 and LepR (a biomarker of BMSCs in bone) on days 3 and 7. LepR$^+$ BMSCs were visible throughout the defects, and there was no significant difference in LepR level between control and PHA-MBG (Fig. 5a, b). In contrast, a significantly higher proportion of BMSCs enriched in the implant of CSN@PHA-MBG compared to control at day 3. Meanwhile, a pronounced reduction in p16 expression on day 3 and 7 was observed in CSN@PHA-MBG compared to control and PHA-MBG groups (Fig. 5a, c) in consistence with the results from flow cytometric analysis (Fig. 3d-i). Moreover, as expected, the defect implanted with CSN@PHA-MBG displayed the lowest proportion of senescent BMSCs (LepR$^+$P16$^+$, Fig. 5a, d). Next, we reconfirmed this trend by flow cytometric analysis with two additional BMSCs markers (CD73$^+$CD90$^+$, Fig. 5e–h). In accordance with the above immunohistochemical results, CSN@PHA-MBG can recruit much more BMSCs (Fig. 5e, f) and meanwhile prevent their senescence locally compared with control and PHA-MBG (Fig. 5g, h).

Bone regeneration is highly related to angiogenesis, and we, therefore, further checked vasculature network formation in the scaffold. We conducted the co-immunostaining of osterix$^+$ osteoprogenitors, which can differentiate into osteoblasts and osteocytes, together with CD31$^+$Emcn$^+$ type H endothelium, which can maintain perivascular osteoprogenitors and couple angiogenesis to

osteogenesis[36]. As shown in Fig. 5i–k, both CD31$^+$Emcn$^+$ type H vessels and associated osterix$^+$ osteoprogenitors were sparse at the defect site of the control group on day 7 and 14. By comparison, the CSN@PHA-MBG scaffold significantly gave rise to the number of osteoprogenitors and type H endothelium at day 7, but PHA-MBG did not (Fig. 5i). Together, it can be concluded that sustained $H_2$ release from CSN@PHA-MBG created a favorable niche for BMSCs recruitment and anti-senescence in great support of angiogenesis and osteogenesis for aging bone defect regeneration.

## The injured aging bone repair outcomes of the hydrogen-releasing scaffold

To further verify the above-mentioned discovery about the SME-remodeling, immune-modulating and regenerability-promoting effects of sustainable $H_2$ supply in vivo and understand their contributions to the therapeutic outcomes, the bone formation rate, tissue morphology, senescent cell burden and anti-osteoclastogenesis potential were comprehensively analyzed.

To evaluate the rate of bone apposition, newly-formed tissue was labeled sequentially with calcein (green fluorescence) at day 11 and alizarin red S (red fluorescence) at day 25. The almost complete red/green fluorescence overlap in the control group indicated a slow bone formation rate and limited self-regeneration ability of aging mice (Fig. 6a). PHA-MBG offered a certain degree of improvement compared with the control group, but there was no significant difference between them (Fig. 6a, b). By comparison, clear sequential fluorescence bands of calcein and alizarin red S suggested that CSN@PHA-MBG significantly enhanced the bone apposition rate (Fig. 6a, b). Further, histological evidence in Fig. 6c demonstrated substantial differences among the control, PHA-MBG and CSN@PHA-MBG groups. In the control group, a large defect gap with obvious fibrous tissue infiltration was observable on day 14, and the ingrowth tissue was mostly fibrous with minimal bone formation after 28 days post operation. In contrast, PHA-MBG and CSN@PHA-MBG treatments showed visibly higher amounts of bone formation. In particular, we observed faster bone formation in the CSN@PHA-MBG treated mice than that of the PHA-MBG group. Especially, the bone defect site of PHA-MBG-treated mice was filled with plentiful chondrocytes on day 14 followed by trabecular bone formation on day 28, and CSN@PHA-MBG induced the formation of increasing new bone (Fig. 6c).

At the end of treatment (day 28), the outcomes of the CSN@PHA-MBG scaffold for aging bone defect repair were further evaluated by synchrotron radiation micro-computed tomography (SRμCT) and immunostaining analyses of neo-tissues. From Fig. 6d, the aging bone defects in the control group without scaffold treatment had not satisfactorily repaired, and even frequently suffered from unpredictable bone fracture. The corresponding quantitative SRμCT analysis exhibited the lowest bone volume/total volume ratio (BV/TV, Fig. 6e), least trabecular number (Tb.N, Fig. 6g) and thinnest trabecular thickness (Tb.Th, Fig. 6f) of repaired bone, confirming the limited innate regeneration capability of aging bone. By contrast, both PHA-MBG and CSN@PHA-MBG improved regenerative outcomes and avoided bone fracture, as revealed by the gradual bony bridging of defects (Fig. 6d). Of note, CSN@PHA-MB with sustained $H_2$-releasing capability considerably improved the repair outcomes of PHA-MBG, as suggested by higher amount of neo-bone formation (BV/TV, Fig. 6e) and better microarchitecture of neo-bone (Fig. 6d, c) after treatment for 28 days.

Moreover, the immunostaining of p16 in neo-tissues (Fig. 6h) revealed pronounced senescent cells (p16$^+$) in both control and PHA-MBG groups at day 28, and no significant difference of p16 contents between control and PHA-MBG groups (Fig. 6i). In contrast, such a senescence phenotype was notably prevented by CSN@PHA-MBG (Fig. 6h, i). Since osteoclastogenesis is positively associated with senescence, we further checked the expression of osteoclastogenic marker Vpp3. Consistently, abundant Vpp3 positivity was detected in

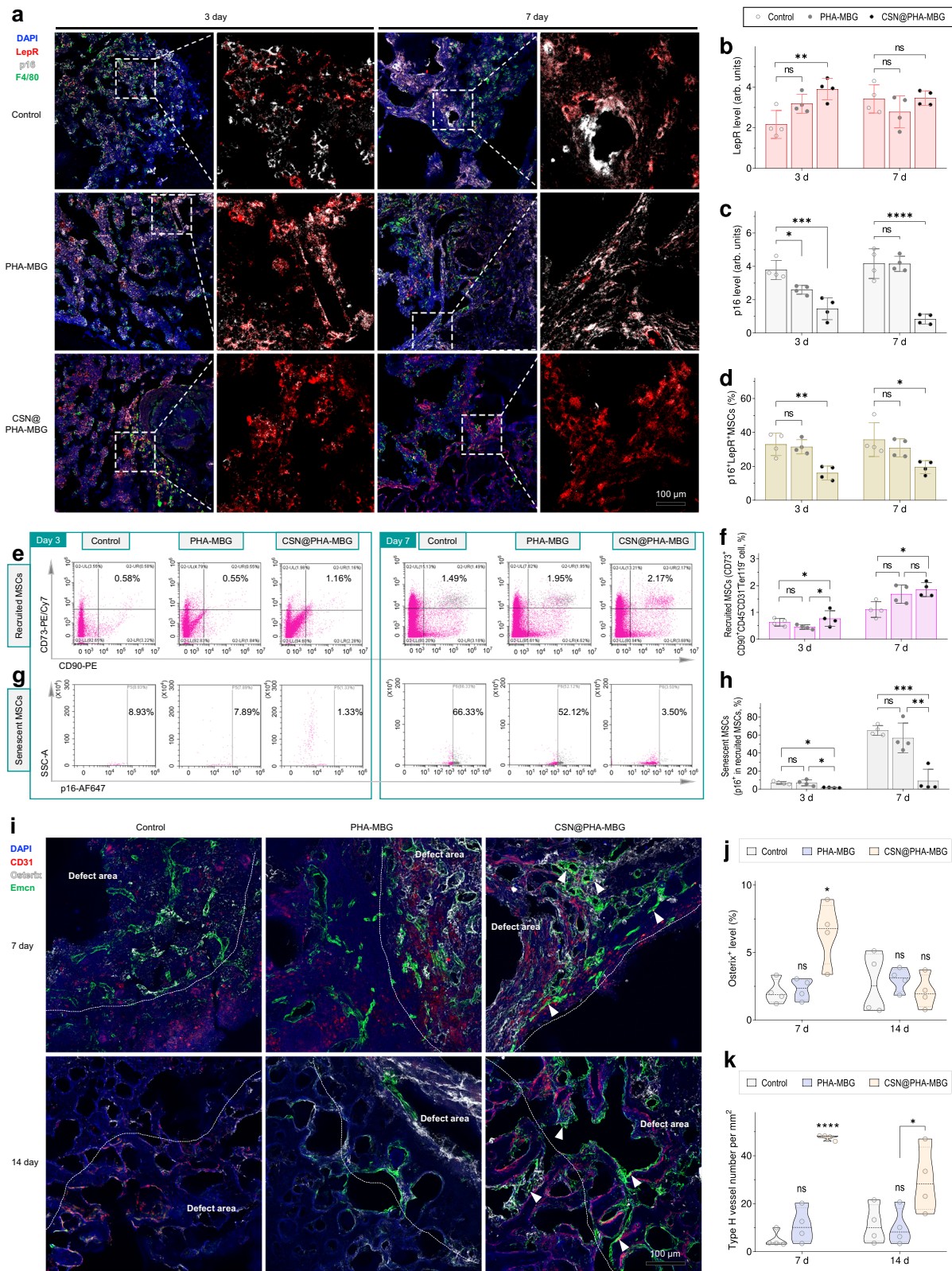

control and PHA-MBG groups, and the percentage of Vpp3-expressing cells in the PHA-MBG group did not differ from the control group. By contrast, an obvious anti-osteoclastogenesis effect was found in the CSN@PHA-MBG group (Fig. 6h, i) in accordance with TRAP measurement results (Supplementary Fig. 31). In addition, no significant differences in osterix expression among these three groups might be because the process of osteoprogenitor recruitment tended to end

after 28-day treatments. These findings, in combination with in vitro osteoclastogenic assay results, pointed out that local $H_2$ release played a predominant role in suppressing osteoclastogenesis via effective SME modeling.

Taken together, we depicted a mechanism illustration, as shown in Fig. 6j. Local and sustained $H_2$ release from the designed scaffold remodeled the SME in time and efficiently by inverting the

**Fig. 5 | The effect of the CSN@PHA-MBG scaffold on the recruitment and senescence of BMSCs in the injured aging bone.** Representative immunostaining images (**a**) and corresponding quantification data (**b**–**d**) of p16 (white) and LepR (red) expression at the aging bone defect site on day 3 and 7 after indicated treatments ($n = 4$ biologically independent samples), representative flow cytometry plots (**e**, **g**) and corresponding quantification analyses (**f**, **h**) of recruited MSCs (**e**, **f**) and senescent MSCs (**g**, **h**) at the defect site on day 3 and 7 after indicated treatments ($n = 4$ biologically independent samples), representative immunostaining images (**i**) and corresponding quantification data (**j**, **k**) of Osterix (white), CD31 (red) and Emcn (green) in the defect site on day 7 and 14 after indicated treatments

($n = 4$ biologically independent samples). In **i**, white triangles indicate the $CD31^+Emcn^+$ type H endothelium. Data are means ± SD. $P$ values in figures **b**–**d**, **f** and **h** were calculated by the one-way ANOVA method with Tukey's post hoc test. **b** $**p = 0.0049$. **c** $***p = 0.0004$, $*p = 0.0253$ in day 3; $****p < 0.0001$ in day 7. **d** $**p = 0.0030$ in day 3; $*p = 0.0218$ in day 7. **f** $*p = 0.0106$, $*p = 0.0145$ in day 3; $*p = 0.0163$ in day 7. **h** $*p = 0.0396$, $*p = 0.0376$ in day 3; $***p = 0.0004$, $**p = 0.0012$ in day 7. $P$ values in **j** and **k** were calculated by Two-tailed unpaired Student's $t$ test. **j** $*p = 0.0121$ in day 7. **k** $****p < 0.0001$ in day 7; $*p = 0.0441$ in day 14; ns means no significant difference ($p > 0.05$). Source data are provided as a Source Data file.

inflammation-extended microenvironment into anti-inflammatory one through macrophage repolarization, and universally attenuating the progression of senescence in various cells. In turn, the SME-remodeling favorable microenvironment promoted the recruitment of MSCs and osteoprogenitors, and locally improved their regenerative capability by reducing their senescent burden, and consequently improved the outcomes of aging bone defect regeneration. In addition, the in vivo biosafety evaluation indicated that both PHA-MBG and CSN@PHA-MBG possessed high biocompatibility without visible damage to major organs (Supplementary Fig. 32) and liver/kidney functions (Supplementary Figs. 33 and 34).

## Discussion

The past decade has witnessed an explosion of understanding cellular senescence in aged organisms and the relationship between cellular senescence and various aging-related pathologies[1,35]. Recently, a threshold theory of senescent cell accumulation has been proposed and emphasized that once senescent cell burden exceeds a threshold, the accumulated senescent cell burden may contribute to tissue destruction and multiple aging-related disorders including immune dysregulation, further amplifying senescence through SASP in a feed-forward loop[37,38]. But moderate senescence can induce the generation of regeneration-associated SASP components such as Ereg in favor of tissue regeneration[38–40], possibly because the degree of senescence is below the threshold. From our present results of SASP components analysis (Fig. 4), the treatment with the hydrogen releasing scaffold has obviously induced the up-regulation of many regeneration-associated SASP components including angiogenesis and osteogenesis factors. Therefore, we here speculate that $H_2$ might reduce the senescent cell burden in aging bone defect to below the threshold by remodeling SME and inhibiting the progress of senescence in various key cells in aging bone, preserving the self-regeneration ability of aging mice. Unfortunately, the definition of senescent cell burden threshold is unclear, the accurate critical degree of senescence is unknown at present, and key components of regeneration-favoring SASP need to be uncovered and scrupulously verified in the future.

The free radical theory regards expressive expression of ROS as a principal cause of oxidative stress, inflammation and aging[41]. Some antioxidants such as vitamin C are proposed to inhibit aging, but their over-administration is prone to break the redox balance of the body and therefore bring some side effects[42,43]. By comparison, $H_2$ has been identified able to selectively scavenge highly oxidative radicals such as •OH under the catalysis of Fe-porphyrin, but rarely affect beneficial radicals, thus exhibiting a wide-spectrum anti-inflammation and anti-senescence effect as well as a high biosafety[12,44,45]. Routine $H_2$ administration routes such as oral uptake of hydrogen-rich water and inhalation of hydrogen gas can neither accumulate a high concentration of $H_2$ at a remote diseased site nor sustain a long time duration of $H_2$ supply. These issues have be well addressed in this work by developing the hydrogen-releasing scaffold which encapsulated CSN as a hydrogen prodrug with a high $H_2$ productivity and a sustained $H_2$ release capability. About one week of $H_2$ release course is able to cover the inflammation and angiogenesis periods of bone regeneration in

support of early-stage SME remodeling in aging bone injury and subsequently improved therapeutic outcomes, but we here envision that a longer $H_2$ release duration, which is able to cover the whole bone repairing progress (over one month) could possibly bring a higher outcome of bone regeneration. It requires the development of more advanced $H_2$ prodrug/scaffolds with a higher sustained release capability in the future.

Moreover, it is worth noting that aging/injury-superposed senescence will provoke immune dysregulation and excessive inflammation, delaying the self-regeneration of injured tissues. Thus, in-time and effective suppression of inflammation at the aging bone defect site is critical important to the SME remodeling for accelerating aging bone repair. The developed hydrogen-releasing scaffold provide a solution to this issue. Notably, SME will polarize senescent macrophages into the pro-inflammatory M1 phenotype, but local sustained $H_2$ release can repolarize senescent macrophages into the anti-inflammatory M2 phenotype, making contribution to creating a pro-regenerative microenvironment for neo-tissue formation. Even though we have confirmed the macrophage-repolarizing effect of $H_2$ in vitro and in vivo in this study, the mechanism of $H_2$ working requires to be disclosed in the further. In addition, we have unexpectedly discovered the effect of $H_2$ on the osteogenic/osteoclastogenic differentiation behaviors of BMSCs and monocytes in SME in this work, but the pathways of $H_2$ influence are unclear at present and need to be investigated in the further.

Currently popular routes to anti-senescence and treatment of senescence-related diseases are administration of senolytic or senomorphic drugs, but both types of anti-senescent drugs lack both anti-senescence selectivity and universality, unavoidably causing obvious toxic side effects[38,46]. In comparison, $H_2$ is an emerging kind of anti-senescence agent totally different from senolytic and senomorphic drugs, and has exhibited high biosafety owing to its anti-oxidation selectivity (only scavenge highly toxic radicals). Moreover, $H_2$ has a specifically high tissue permeability[47–51], which allows $H_2$ to easily penetrate bone tissue to play the SME-remodeling function. Further, the developed CSN@PHA-MBG scaffold offers a promising solution to realize a high-dosage, local and sustained $H_2$ delivery in support of high efficacy of aging bone defect repair. In addition, the developed CSN@PHA-MBG scaffold exhibits high biocompatibility, and its degradation products involving $Ca^{2+}$, $SiO_3^{2-}$, $H_2$ and 3-hydroxyalkanoates, have been widely demonstrated to be highly biocompatible, lowly toxic, easily metabolic in the body and also osteoinductive for bone regeneration[31,32]. High biosafety and therapeutic validity endow the CSN@PHA-MBG scaffold with a high potential for clinical translation.

Carlock et al. found that the clinical outcomes of autogenous iliac crest bone graft (ICBG) for nonunion repair appeared to be as effective in elderly patients as in the young with a long bone nonunion, and ICBG is considered as the benchmark graft for nonunion repair among all age groups[52,53]. We speculate that high biocompatibility and osteoconduction of ICBG might bring a minimum burden for surrounding senescent cells/tissues compared with allotransplantation and artificial graft so that the accumulated senescent cell burden would not excess the threshold for self-

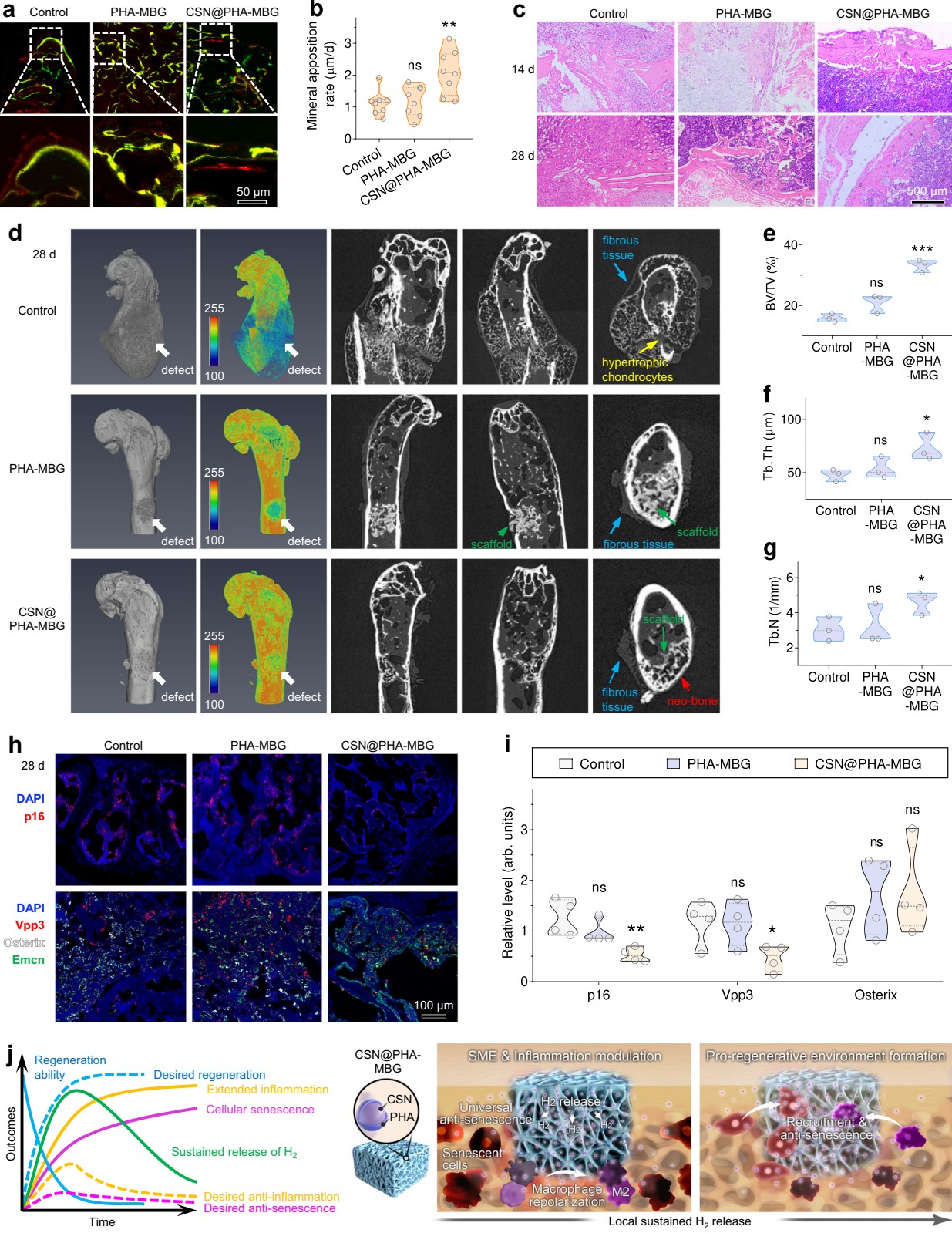

regenation according to the above threshold theory. But when ICBG is unavailable, the graft of scaffold is also an important candidate. In this case, the effect of aging on critical-size bone defect repair should not be negligible, and local hydrogen supply seems favorable for accelerating aging bone repair from present experimental results, which needs further clinical confirmation.

## Methods

### Primary cells isolation, culture and characterization

All experiment protocols were evaluated and approved by Institutional Animal Care and Use Committee (SIAT-IACUC-200320-YGS-TW-A1184) of Shenzhen Institute of Advanced Technology, Chinese Academy of Sciences. C57BL/6 J male mice were purchased from Beijing Vital River

**Fig. 6 | The outcomes of aging bone defect repair with the CSN@PHA-MBG scaffold.** Sequential fluorescence labelings of bone formation with calcein (green) and alizarin red S (red) (**a**) and the corresponding quantitative analysis of mineral apposition rate (**b**) ($n = 8$ biologically independent samples), histological examination of neo-tissues at the aging bone defect site after indicated treatments on day 14 and 28 (**c**), representative SRμCT images of neo-tissues at day 28 after control, PHA-MBG or CSN@PHA-MBG treatment (**d**) and the corresponding quantification of BV/TV (**e**), Tb.Th (**f**), and Tb.N (**g**) ($n = 3$ biologically independent samples), representative immunostaining images (**h**) and the corresponding quantification data (**i**) of p16 (red), Vpp3 (red), Osterix (white) and Emcn (green) expressions in neo-tissues on day 28 after indicated treatments ($n = 4$ biologically

independent samples), illustration of the mechanism of hydrogen-mediated injured aging bone regeneration, where local and sustained $H_2$ release from the CSN@PHA-MBG scaffold remodeled the SME by inducing effective anti-inflammation (macrophage polarization) and universal anti-senescence, and consequently promoted MSCs recruitment and preserved their regenerative capability (**j**). BV/TV, the ratio of bone volume to total volume; Tb.N, trabecular number; Tb.Th, trabecular thickness. Data are means ± SD. *P* values were calculated by the Two-tailed unpaired Student's *t* test method. **b** **$p = 0.0040$; **e** ***$p = 0.0003$; **f** *$p = 0.0370$; **g** *$p = 0.0481$; **i** **$p = 0.0086$ for p16, *$p = 0.0317$ for Vpp3; ns means no significant difference ($p > 0.05$). Source data are provided as a Source Data file.

Laboratory Animal Technology Co., Ltd. and kept under a 12/12 h light/dark cycle, 24–26 °C, and 60% humidity.

Bone marrow mesenchymal stem cells (BMSCs) were extracted following established protocols[54]. Firstly, the 24-month-old C57BL/6 J male mice were mercifully killed by cervical dislocation and immersed in 75% alcohol for 5 min to sterilize the body surface. The hind limbs were then extracted and the femurs and tibiae were isolated. After removing the attached soft tissue, all the epiphyses were cut off and fresh bone marrow was collected by flushing the bone marrow cavity three times with the HBSS buffer (Ca/Mg-free, Gibco, 14170120) containing 2% FBS (Gibco, 10099141 C), followed by centrifugation for 10 min at 300 × *g*. After discard of the supernatant, the cell pellet was resuspended and digested in 2 mL of the HBSS buffer containing 3 mg mL⁻¹ of type I collagenase (Gibco 17018029), 4 mg mL⁻¹ of neutral protease (Roche, 4942078001), and 200 U mL⁻¹ of DNAase (Thermo, 89836) at 37 °C for 15 min. The digestion was stopped by adding 10 mL of the HBSS (Mg/Ca-free, Gibco, 14170120) containing 2% FBS and 2 μM of EDTA (Invitrogen, AM9260G). Next, cells were collected by centrifugation at 300 × *g* for 10 min, and then resuspended in the HBSS buffer (Ca/Mg-free) containing 2% FBS. After filtering with a 40 μm strainer, digested single cells were collected via centrifugation and cultured in the α-minimal essential medium (α-MEM, Hyclone, SH30265.01) containing 20% FBS, 1% sodium pyruvate (Gibco, 11360070) and 10 μM Rock inhibitor (MedChemExpress, HY-10583). At 12 h after incubation, nonadherent cells were washed away with DPBS (Gibco, 14190144), and adherent BMSCs were cultured in the above fresh α-minimal essential medium. The BMSCs proliferation was determined by the CCK-8 method (Beyotime, C0046). Briefly, BMSCs were seeded at a density of $2 \times 10^4$ cells/well in the 24-well plate (Corning, 3524). After 1, 3, 7 days of culture, the culture medium was replaced by a 10% CCK-8 solution (vol/vol) and incubated for 2 h. The optical density (OD) value at 450 nm was measured by a microplate reader (TECAN, Infinite® 200 PRO), and the data were normalized to the OD value of 1 day.

Osteocytes were isolated from the femurs and tibiae according to a previously reported method[55]. As mentioned above, the femurs and tibiae of 24-month-old C57BL/6 J male mice were collected, and then cut into tiny pieces which were immediately digested three times with collagenase (2.5 mg mL⁻¹ in α-MEM, Sigma, C9891) under shaking at 37 °C for 30 min per time. At each digestion interval, bone pieces were washed twice with HBSS (Mg/Ca-free). Afterward, bone pieces were digested with 5 mM EDTA solution (in PBS containing 0.1% BSA) and collagenase solution alternately for three times, as demonstrated in Fig. S1. The cells after the digestion of step 4 were collected, seeded on rat tail type I collagen (Corning, 354236) and cultured in the α-MEM containing 10% FBS.

Macrophages were differentiated from monocytes isolated from fresh bone marrow according to previously reported methods[56]. Briefly, fresh bone marrow was flushed out with the ice-cold PBS containing 2% FBS from the femurs and tibiae collected from 24-month-old C57BL/6 J male mice as mentioned. After treatment with red blood cell lysis buffer (Gibco, A1049201), monocytes were isolated via density gradient centrifugation with Ficoll (GE Healthcare, 17144002).

To obtain the fully differentiated macrophages, the monocytes were differentiated in the 30% L929-conditioned medium (in DMEM) containing 20% FBS for 6 d. For macrophage phenotype characterization, macrophages ($5 \times 10^4$ cells/well, from 24-month-old mice) were seeded in 24-well plates and incubated with following treatments: control (in the general incubator), $H_2$ treatment (in the hydrogen incubator), LPS stimulation (100 ng mL⁻¹, Sigma, SMB00610) and IL-4 stimulation (10 ng mL⁻¹, Peprotech, 214-14). After 3-day or 7-day treatment, the production of cytokines in macrophage was analyzed using TNF-α (EMC102a, NeoBioscience) and TGF-β1 (EMC107b, NeoBioscience) ELISA kits.

For the hydrogen culture condition, cells were incubated at 37 °C in a humidified atmosphere containing 60% $H_2$, 14% $N_2$, 21% $O_2$ and 5% $CO_2$ in a hydrogen incubator (Puhe, Wuxi). Cells incubated in the standard atmosphere (95% air and 5% $CO_2$) were used as control. Cell culture medium was replaced with a hydrogen-saturated fresh one once every 2 days, which was prepared by blowing hydrogen gas into fresh culture medium for 30 min. Different concentrations of hydrogen (25, 50, 60%) for only replacing nitrogen in the hydrogen incubator were set to monitor the hydrogen concentration in the culture medium, and it was found from Supplementary Fig. 35 that the hydrogen contents in the culture medium achieved the saturation value within the almost same time period of about 30 min, possibly owing to a high flow volume of hydrogen gas in the hydrogen incubator. Even though it was almost impossible to fully simulate the in vivo hydrogen release behavior of the CSN@PHA-MBG scaffold (Supplementary Fig. 20) using the hydrogen incubator (Supplementary Fig. 35) and the scaffold cannot achieve the saturated concentration of $H_2$ in vivo as the hydrogen incubator did in vitro, the designed scaffold can maintain a long $H_2$ release time duration to increase the dose of $H_2$. Therefore, the results from experiments using the hydrogen incubator only attempted to suggest the necessity of the sustained supply of $H_2$ for increasing the anti-senescence outcome.

## Intracellular ROS analysis

The 2,7-dichlorodihydrofluorescein diacetate (DCFH-DA, Beyotime, S0033) assay was used for the detection of ROS. After 7 days of cell culture, 1 mL of DCFH-DA working solution (20 μM, in FBS-free cell culture media) was added to each group and allowed to react for 30 min at 37 °C. After counterstaining with 4′,6-diamidino-2-phenylindole (DAPI, Beyotime, C1006) for 15 min, cells were washed and imaged by confocal laser scanning microscope (CLSM, Leica, SP5). Fluorescence intensity was analyzed using ImageJ software.

## Immunofluorescence staining

For osteocyte characterization, the harvested osteocytes were labeled with SOST (Bioss, bs-10200R, 1:200). For phenotypic analysis, macrophages were labeled with CD86 (Sant Cruz, sc-28347, 1:400) and CD206 (CST, 24595 S, 1:400). For senescence analysis, immunofluorescence staining of HO-1 (Abcam, ab68477, 1:500), p16 (Abcam, ab211542, 1:500), Ki67 (Novus, NB500-170, 1:300), γ-H2A.X (CST, 9718 T, 1:300) and p21 (Abcam, ab188224, 1:1000) was performed. Briefly, cells were fixed with 4% paraformaldehyde (Servicebio, G1101-

500 mL) for 15 min at 4 °C. After washed with PBS, the fixed cells were permeabilized with 0.1% Triton X-100 for 15 min, and then blocked with 5% goat serum for 1 h at room temperature. Subsequently, cells were incubated with specific primary antibody at 4 °C overnight. Next, cells were washed with PBS and incubated with Alexa Fluor-coupled secondary antibody for 1 h at room temperature. Thereafter, nuclei were counterstained with DAPI and imaged with CLSM. The following secondary antibody was used: Alexa Fluor 647 conjugated goat anti-mouse IgG H&L (Abcam, ab150115, 1:400) and Alexa Fluor 488 conjugated goat anti-rabbit IgG H&L (Abcam, ab150081, 1:250). Quantitative assessment of fluorescence intensity in cells was analyzed with ImageJ software.

### Senescence-associated β-galactosidase (SA-β-gal) assay
SA-β-gal staining was used as a general marker of cellular senescence. Briefly, cells were fixed with 4% formaldehyde and washed three times with PBS. Then, the fixed cells were stained with a SA-β-gal staining solution (Beyotime, C0602) at pH 6.0 and incubated at 37 °C overnight in dark. After washing with deionized water, the SA-β-gal stained cells were observed microscopically.

### FISH measurement
FISH technology was used for TAFs analysis according to the manufacturer. Briefly, cells were fixed with 4% formaldehyde for 4 min at 37 °C. After washing with PBS, cells were treated with $100 \, \mu g \, mL^{-1}$ RNase A (ST576, Beyotime) in 2X SSC (ST462, Beyotime) for 1 h at 37 °C, and then with a pepsin solution (50 µL 5% pepsin stock (P6887, Sigma) in 50 mL 0.01 M HCl) for 4 min at 37 °C. Subsequently, cells were fixed with 4% formaldehyde for another 4 min at 37 °C again, washed with PBS, and then dehydrated for 1 min in 70%, 85%, 100% cold ethanol solutions orderly. Next, cells were dried at 80 °C for 5 min, and then activated at 85 °C for 15 min in a PNA probe (F1003, Panagene, 200 nM) hybridization solution (20 mM $Na_2HPO_4$, pH 7.4, 20 mM Tris, 60% formamide, $0.1 \, \mu g \, mL^{-1}$ salmon sperm DNA, 2X SSC), followed by further incubation in dark at room temperature for 1 h. After washing with 0.1% Tween-20 (in 2X SSC) for 10 min at 55–60 °C, γ-H2A.X (9718 T, CST, 1:250) and DAPI (D3571, Invitrogen) were stained.

### Measurement of the senomorphic and senolytic behaviors of $H_2$
Briefly, osteocytes were extracted from the bone of 24-month-old aging mice, and then incubated with $H_2$ in the $H_2$ incubator, or with a typical senomorphic agent (ruxolitinib, 1 µM, AbMole, M1787) in the general incubator without $H_2$ supplement, or with a typical senolytic agent (5 µM quercetin (AbMole, M3902) plus 1 µM dasatinib (AbMole, M1701)) in the general incubator without $H_2$ supplement. After treatment for 3 days, a SA-β-Gal kit (C0602, Beyotime) and a CCK-8 kit (C0037, Beyotime) were used to measure SA-β-gal expression and cell viability, respectively.

### Gene-expression analysis
The gene expression levels were determined by quantitative reverse-transcription PCR (qRT-PCR). Total RNA was extracted using a Trizol reagent (TAKARA, 9109). RNA concentration was quantified by measuring the absorbance at 260 nm using NanoQuant (TECAN, Infinite® 200 PRO). Complementary DNA was synthesized from 2 µg of RNA by a PrimeScript RT reagent kit (Takara, RR037A) following manufacturer's instructions. RT-PCR assay was performed by Thermocycler system (Biometra) using SYBR Premix Ex Taq Kit (Takara, RR420A). Cycle conditions were demonstrated as follows: denaturation at 95 °C for 30 s; 40 cycles at 95 °C for 5 s and 60 °C for 30 s; melt curve from 65 °C to 95 °C with an increment of 0.5 °C/5 s. Osteocytes markers (*E11/gp38*, and *Sost*), inflammatory markers (*iNOS*, *Arg1*, and *Il-10*), senescence-associated markers (*p16*, *p21*, and *p53*) and osteoblastogenic markers (*Alp, Runx2, Osterix, Ocn, Col.I,* and *Opn*) were evaluated. *Gapdh* or *β-actin* gene was used as a house keeping gene. The data

were analyzed using the comparative Ct ($2^{-\Delta\Delta Ct}$) method and expressed as a fold change with respect to the control. Primer sequences used in this study were listed in Supplementary Table 1.

### In vitro osteoclast differentiation and TRAP analysis
Monocytes were harvested from bone marrow and pre-cultured for 24 h in vehicle (negative control, α-MEM complete medium), $25 \, ng \, mL^{-1}$ M-CSF (positive control, α-MEM complete medium containing $25 \, ng \, mL^{-1}$ M-CSF (Peprotech, 315-02)), conditioned medium from senescent cells (α-MEM complete medium containing 50% supernatant form senescent BMSCs), or conditioned medium from $H_2$-treated senescent BMSCs (α-MEM complete medium containing 50% supernatant form $H_2$-treated senescent BMSCs), followed by induction in the osteoclast differentiation medium (α-MEM complete medium suppled with $200 \, ng \, mL^{-1}$ RANKL (Peprotech, 315-11) and $25 \, ng \, mL^{-1}$ M-CSF) for 3 d. TRAP staining (Solarbio, G1492) and TRAP enzyme activity (Solarbio, BC5405) were performed according to the manufacturer's instructions.

### In vitro osteogenic differentiation and quantification
The BMSCs were plated at a density of $1 \times 10^5$ cells per well of 24-well plate (Corning, 3524). After reaching confluency, the medium was replaced by an osteogenic medium (α-MEM, 10% FBS, $50 \, \mu g \, mL^{-1}$ ascorbic acid, 100 nM dexamethasone and 10 mM β-glycerophosphate). Differentiation medium was refreshed every 3 d.

For mineralization assay, at day 21, cells were fixed with 4% paraformaldehyde, washed twice with ultrapure water and stained with a 1% alizarin red staining solution (ARS, pH 4.2, sigma-Aldrich, A5533-25G) for 30 min at 37 °C. After washing 5 times with ultrapure water, the staining was imaged using microscopy. For quantification, ARS was de-stained with 10% cetrlpyridinium chloride for 1 h, and the absorbance at 540 nm was read by microplate reader. ARS absorbance was normalized by total protein content. The total protein content was measured by following the manufacturer's instructions using the BCA protein assay kit (Beyotime, P0010S).

### Preparation of mesoporous bioactive glass (MBG) scaffold
MBG scaffolds were prepared according to our previous report[57]. Briefly, 4.0 g of F127 (Sigma, P2443), 0.76 g of $Ca(NO_3)_2 \cdot 4H_2O$ (Aladdin, C100072) and 0.23 g of TEP (Aladdin, T103502) were successively dissolved in 50 mL ethanol containing 1.0 g of 0.5 M HCl. 5.2 g of TEOS (Aladdin, T110596) was then added to the above solution and vigorously stirred for another 24 h. After rotary evaporation at 60 °C for 30 min, the resulting viscous MBG sol was uniformly impregnated on customized polyurethane sponges. The final MBG scaffolds were obtained via calcination at 600 °C for 6 h.

### Preparation and characterization of CSN
CSN was synthesized by the microwave-assisted mechanical stripping method using ethylene glycol (Macklin, E808735-500mL) as a dispersant. Firstly, 100 mg of $Ca_2Si$ raw powder (25 µm, 99.9%, Jinzhou Haixin Metal Technology Co., LTD), 400 mg of poly(vinyl pyrrolidone) (PVP, Sigma, 856568), 140 mg of EDTA disodium salt (EDTA-2Na, Macklin, E809185) and 100 mg of hyaluronic acid sodium (Macklin, H6350) were completely dispersed in 20 mL ethylene glycol under vigorous stirring. The mixture was then transferred to a 20 mL autoclave and heated at 160 °C for 3 h using a microwave synthesizer (400 W, UWAVE-2000). After cooling to 50 °C, the product was crushed in a pulsed sonicator ($425 \, W \, cm^{-2}$, 6.6 s on/2.2 s off, Shanghai Hanuo) for 90 min. To remove the big particles, the product was centrifuged at 4000 rpm for 10 min and the supernatant was collected. After washing three times with ethanol/water (1:1, vol/vol) mixture, the final product was obtained by centrifugation at 12,000 rpm for 20 min, and redispersed in 20 mL of ethanol. The morphology, elemental mapping, and crystal structure of CSN were

analyzed by SEM, energy-dispersive spectroscopy (EDS, Thermo, APREO S) and powder X-ray diffraction (XRD, Rigaku, MiniFlex600), respectively.

## Synthesis and characterization of CSN@PHA-MBG scaffolds

The scaffolds with the size of Ø10 ×3 mm and Ø1.2 × 2 mm were used for in vitro or in vivo studies, respectively. The CSN@PHA-MBG scaffolds were prepared by an electrostatic spraying strategy. Taking the CSN@PHA-MBG scaffold (Ø10 × 3 mm) containing 50 µg of CSN as an example, the preparation of CSN@PHA-MBG was conducted as follows: 1.5 mg of poly(3-hydroxybutyrate-*co*-3-hydroxyvalerate-*co*-3-hydroxyhexanoate) (PHA, Mw = 25,000 Da, Bluepha) was dissolved in 10 mL of DCM/DMF (8:2, vol/vol) solution. Next, CSN was uniformly mixed with the PHA solution at a particle concentration of 4 mg mL$^{-1}$. After dispersion by sonication, the CSN@PHA mixture was transferred into a 5 mL syringe and sprayed onto a MBG scaffold through a 22-guage needle at a rate of 0.5 mL h$^{-1}$ for 3 h. An 18 kV voltage was applied to create an electrostatic field within 10 cm of working distance. To achieve a desired spraying outcome, the MBG scaffold was kept turning. The CSN@PHA-MBG scaffolds with different CSN contents were prepared under the identical conditions except the CSN concentration was adjusted. The composition of CSN@PHA was analyzed by EDS mapping. The morphology of CSN@PHA and CSN@PHA-MBG was characterized by SEM (Zeiss Supra55). The biocompatibility of PHA-MBG and CSN@PHA-MBG was evaluated by Live/Dead (Thermo, L3224) assay. For in vitro and in vivo experiments, all scaffolds were sterilized by gamma irradiation (Shenzhen Jinpengyuan Irradiation Technology Co., Ltd.) at 15 kGy for 10 h.

## Hydrogen release behavior of CSN and CSN@PHA-MBG in vitro and in vivo

Hydrogen release behaviors of CSN and CSN@PHA-MBG in the PBS buffer (pH 7.4) were detected by a hydrogen microelectrode (Unisense, Denmark) or GC (Agilent Technologies, 7890B), and hydrogen concentration was calculated based on a pre-established hydrogen calibration curve with gradient concentrations of hydrogen-dissolved water (Supplementary Fig. 36).

For in vivo hydrogen release measurement, after implantation of the CSN@PHA-MBG scaffold into the bone defect site for different time periods (1, 3, 5, 7, 9, 11 days), the whole joint at the site of defect was extracted and immediately placed in a sealed penicillin bottle pre-filled with PBS (pH = 7.4) for 2-min grinding, followed by the measurement of hydrogen concentration by GC.

## Surgical procedures

A mouse femoral plug defect model of critical size bone defect was established in this study. C57BL/6 male mice (24 months old) were randomly divided into three groups ($n = 10$ at each time point) as follows: control (defect without an implant), PHA-MBG, and CSN@PHA-MBG. Briefly, after anesthetization with isoflurane (RWD, R510-22-10), an 8 mm-long straight incision was made in the distal femoral epiphysis of each hind leg, and then a 1.2 mm-diameter round defect was created perpendicularly in the distal region of the femur diaphysis by using a trephine bur. After the defect was rinsed with saline solution, a cylinder scaffold was implanted into the defect, and the incision was closed using silk suture (Jinhuan Medical, C412). To prevent from infection, all mice were provided with drinking water containing sulfamethoxazole (15 mg/kg, Merck, PHR1126-1G) and trimethoprim (30 mg/kg, Merck, PHR1056-1G) for 7 days after surgery. To relieve the pain, all mice were orally administrated with ibuprofen (10 mg/kg, Merck, PHR1004-1G) in drinking water for 24 h after surgry. The estrogen cycle of female mice will significantly inhibit bone growth and tend to cause osteoporosis, which will lead to an obvious difference of aging bone repair between male and female. Therefore, in order to

avoid the aberration and enhance the comparability, only male mice were used for the bone defect model in this work.

## Flow cytometry

For flow cytometric analysis, fresh bone specimens were collected, and soft tissues attached onto the bone were removed. To obtain single cell suspensions, each specimen was crushed in the ice-cold Ca/Mg-free HBSS buffer containing 2% FBS with mortar and pestle, followed by enzymatic digestion with HBSS buffer containing collagenase (3 mg mL$^{-1}$), neutral protease (4 mg mL$^{-1}$), and DNAse (200 U mL$^{-1}$) at 37 °C for 15 min. The digestion was quenched by addition of the HBSS buffer (Mg/Ca-free) containing 2% FBS and 2 µM of EDTA. After filtering using a 40 µm strainer (Corning, 352340), cells were collected via centrifugation and resuspended in PBS. Then, the cells were counted and adjusted to the density of $1 \times 10^6$ cells/100 µL in PBS. To determine cell viability, all the cells were labeled with a near-infrared fluorescent reactive dye (Thermo, L34976, 1:1000) at 4 °C for 30 min. After washing with PBS, the cells were blocked with anti-mouse CD16/32 antibody (Biolegend, 101320, 1:50) at 4 °C for 10 min. Then, an equal number of cells (density of $1 \times 10^6$ cells/100 µL) were subjected to immunostaining.

For the analysis of senescent immune cells, cells were stained with FITC-conjugated CD45 (BD, 553079, 1:1000), PerCP-Cy5.5 conjugated Ly6G (BD, 560602, 1:100), BV421 conjugated CD11b (BD, 562605, 1:1000), PE conjugated F4/80 (BD, 565410, 1:800) antibodies at 4 °C for 45 min prior to intracellular staining. Then, cells were fixed and permeabilized using a transcription factor buffer set (BD, 562574), and probed with anti-p16 antibody (Abcam, ab211542, 1:200) at 4 °C for 45 min. After washing, cells were incubated with Alexa Fluor 647 conjugated secondary antibody (Abcam, ab150075, 1:500) at 4 °C for 45 min.

For the analysis of macrophage polarization, cells were stained with FITC-conjugated CD45, PerCP-Cy5.5 conjugated Ly6G, BV421 conjugated CD11b, PE conjugated F4/80, and PE-Cy7 conjugated CD86 (BD, 560582, 1:100) antibodies at 4 °C for 45 min. After fixation and permeabilization (Thermo, 88-8824), cells were incubated with Alexa Fluor 647 conjugated CD206 antibody (BD, 565250, 1:80) at 4 °C for 45 min.

For the analysis of MSCs, cells were stained with FITC-conjugated CD45, BV421 conjugated CD31 (Biolegend, 102424, 1:200), BV605 conjugated Ter119 (Biolegend, 116239, 1:200), PE/Cyanine7 conjugated CD73 (Biolegend, 127224, 1:160), and PE conjugated CD90 (BD, 553006, 1:200) antibodies at 4 °C for 45 min. For the analysis of senescent MSCs, cells were then fixed, permeabilized, and probed with primary p16 antibody at 4 °C for 45 min, followed by further incubation with Alexa Fluor 647 conjugated secondary antibody at 4 °C for 45 min.

The isotype-matched immunoglobulins were used as isotype controls under the same conditions. The gates were set according to the isotype controls. All the steps were performed according to the manufacturer's instructions. Flow cytometry of cells was assessed and analyzed using a CytoFLEX-S cytometer (Beckman Coulter).

## Quantification of in vivo cytokines

To collect the implant supernatant, implants were harvested, grinded and centrifuged at 300 × g at 4 °C for 15 min. The levels of cytokines in the supernatant were analyzed using a proteome profiler mouse cytokine array kit (R&D systems, ARY028), and quantified by a series of mouse ELISA kits: IL-4 (Neobioscience, EMC003.96), IL-10 (Neobioscience, EMC005.96), IL-6 (Neobioscience, EMC004.96), IL-1β (Neobioscience, EMC001b.96), IL-13 (Jianglai biology, JL20247), endoglin (Jianglai biology, JL28764), osteopontin (Jianglai biology, JL10068), and osteoprotegerin (Jianglai biology, JL10936).

## SRµCT measurement

Femora were retrieved from mice, fixed overnight in 4% paraformaldehyde and analyzed by SRµCT. The SRµCT measurement was

carried out at the BL13HB beamline of the Shanghai Synchrotron Radiation Facility (Shanghai, China) using a monochromatic beam with energy of 20 keV and a sample-to-detector distance of 0.2 m. A 2048 × 2048 sCMOS detector (Flash 4.0, Hamamatsu City, Japan) with the pixel size set to 5.2 μm was used to record tomographies. An Avizo software was used to visualize the 3D structure of slices. The trabecular bone volume fraction (BV/TV), trabecular thickness (Tb. Th), and trabecular number (Tb. N) of the trabecular bone region of interest were measured using a Brucker software.

### Histology and immunofluorescence analysis

For histological analysis, retrieved femora were fixed in a 4% neutral paraformaldehyde buffer for 72 h, dehydrated, embedded, and sectioned (Leica, RM2265) at a 4.5 μm thickness. Sections were stained with hematoxylin and eosin (H&E, Servicebio, G1076-500mL) to examine their histological morphology.

To examine dynamic bone formation between week 2 and week 4, we subcutaneously injected 0.1% calcein (Sigma, C0875, 10 mg kg$^{-1}$ body weight) and 0.15% Alizarin Red S (Sigma, A5533, 15 mg kg$^{-1}$ body weight) in PBS into mice at day 17 and 3 prior to euthanasia after operation for 4 weeks, respectively. Then, the specimens were cut, ground and polished into a final thickness of 40 μm. The undecalcified sections were observed using CLSM. The excitation/emission wavelengths of calcein and Alizarin Red S were 488/517 nm and 543/617 nm, respectively.

For immunofluorescence analysis, fresh tissue samples were immediately fixed in an ice-cold 4% paraformaldehyde solution for 4 h at 4 °C. After immersing into 20% sucrose and 2% polyvinylpyrrolidone overnight, all samples were embedded in an OCT compound (Leica) and cut into sections of 10 μm or 40 μm (Leica, CM1950) in thickness. Immunofluorescence staining and analysis were performed as follows. Briefly, after treatment with 0.3% Triton X-100 for 20 min, tissue sections were blocked with 5% goat serum at room temperature for 1 h. Afterward, we incubated tissue sections with individual primary antibodies to mouse CD31 (Abcam, ab28364, 1:100), Endomucin (Santa Cruz, sc-65495, 1:100), Osterix (Santa Cruz, sc-22536-R, 1:100), Leptin R (R&D system, BAF497, 1:200), Vpp3 (Abcam, ab200839, 1:100), F4/80 (Abcam, ab6640, 1:100), iNOs (Abcam, ab15323, 1:100), Arg1 (NOVUS Biologicals, NBP2-03618, 1:100), and p16 (CST, 29271 S, 1:100) overnight at 4 °C. Primary antibodies were then visualized with following species-appropriate secondary antibodies: Alexa Fluor 555 conjugated goat anti-rabbit IgG H&L (Abcam, ab150086, 1:400), Alexa Fluor 555 conjugated streptavidin (Thermo, S32355, 1:400), Alexa Fluor 488 conjugated goat anti-rat IgG H&L (Jackson ImmunoResearch, 112-545-062, 1:400) and Alexa Fluor 647 conjugated goat anti-mouse IgG H&L (Jackson ImmunoResearch, 115-605-003, 1:400). The sections were mounted by anti-fade reagent with DAPI (CST, 8961 S). A Leica confocal microscope was used to image samples.

### Biochemical measurement

After 28 days of implantation, 1 mL of the whole blood was collected from the orbital vein of isoflurane anesthetized mice and analyzed by a cell analyzer (BC-31S, Mindray). For analysis of liver and kidney functions, the blood was centrifuged at 2000 × g at 4 °C for 15 min and the supernatant was then measured on a biochemical analyzer (iMagic-M7).

### Statistical analysis

Statistical analysis was performed with GraphPad Prism 9 (GraphPad Software). For comparison of two groups, two-tailed Student's $t$ tests were used. For comparison of multiple experimental groups, either one-way analysis of variance (ANOVA) or two-way ANOVA method was used where indicated. The value of $p < 0.05$ was considered significant. *$p < 0.05$, **$p < 0.01$, ***$p < 0.001$, ****$p < 0.0001$; ns not significant.

### Reporting summary

Further information on research design is available in the Nature Portfolio Reporting Summary linked to this article.

### Data availability

All the data supporting the findings of this study are available within the article and its Supplementary Information files. Any additional requests for information can be directed to, and will be fulfilled by, the corresponding authors. Source data are provided with this paper.

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

## Acknowledgements

The authors greatly appreciate the help of Lin Tang (Chongqing University) and Yun Ou (Hunan University of Science and Technology) in the electrospraying experiment, and the Instrumental Analysis Center of Shenzhen University (XiLi campus) for assistance in material characterizations. This work was supported by the National Natural Science Foundation of China (32071346, 32271403, 51872188, 81927804, and 82172078), the NSFC Science Center Program (T2288102), Natural Science Foundation of Guangdong Province (2021A1515011155), Shenzhen Science and Technology Program (JCYJ20190807164803603, RCJC20210706092010008), Shenzhen Basic Research Program (SGDX20201103093600004), Youth Innovation Promotion Association of the Chinese Academy of Sciences (2021364), the National Postdoctoral Program for Innovative Talents (BX2021101), and Center of Hydrogen Science, Shanghai Jiao Tong University, China.

## Author contributions

W.T., C.S.L., and Q.J.H. proposed the concept and designed the project. W.T. and Q.J.H. wrote the manuscript. S.Q.C., S.Q.X., and W.T. performed experiments under the assistance of D.N.L., W.S., and S.Z.C. Y.M.Y. performed the SRμCT measurement, histology and immunofluorescence analysis of in vivo investigation. G.L.L., W.T., C.S.L., and Q.J.H. analyzed and interpreted the data.

## Competing interests

The authors declare no competing interests.
