## [Peer Review File · Nature Communications]

REVIEWER COMMENTS

Reviewer #1 (Remarks to the Author):

The authors found that sustainable H2 supply has the potential to remodeling the senescence microenvironment through unique and superior technology. Because H2 exerts various effects in most organs, we predicted a role in the slowdown of the aging process. The authors clearly demonstrated the H2 effect on anti-senescence by providing a large number of convincing experimental results. The technology is original and wonderful.

This work definitely excels in our field. There are few publications related to H2 on the role of H2 in aging. The results are not simply descriptive, but satisfying a wide range of readers.

This study fully supports the conclusions and claims. No additional experiments are required.

There are no flaws in the analysis, interpretation or conclusions of the data. There is no reason to forbid publication and no revision is necessary.

This work exceeds the standards expected in our field.

Enough details are provided for anyone to reproduce the work.

I have no additional remarks.

Reviewer #2 (Remarks to the Author):

The authors have addressed many of my previous concerns from the Nature Aging review. However, I am still concerned that their phenotyping of senescence in vivo is incomplete. They do not provide any in vivo evidence for definitive markers of senescence (eg, telomere-associated foci or senescence-associated distension of satellite DNA). In most of the paper, they monitor senescence simply by expression of p16 - and this monitoring by flow appears to be still by a very questionable and unvalidated Santa Cruz p16 antibody (see below).

It would help if they could provide some more direct link, even if in vitro, between H2 and senescence. Specifically, do they have any evidence that H2 kills senescent but not non-senescent cells (ie, is senolytic). Or is H2 simply anti-inflammatory - and thus is functioning more as a "senomorphic" agent and reducing the SASP of senescent cells without killing them. And are the effects of H2 specific (or relatively specific) for senescent cells, or is this just a general anti-inflammatory action of H2. I believe the link to senescence is the weakest part of this paper and should be addressed in more detail.

Another significant issue is that they appear to use 3 different antibodies for p16 (Abcam on page 22, still a Santa Cruz - not validated - for flow (page 27) and Cell Signaling on page 28). This does make the critical flow data less credible and remains a significant concern.

Reviewer #4 (Remarks to the Author):

The manuscript by Chen et al. describes the efficacy of H2 (released from a biomaterial) to improve the SME, and hence regenerative capacity. The work seems interesting, but some issues apply:

Title

1) the words 'sustained' and 'aging' are preferable to be omitted from the title given the lack thereof in the described work

Abstract

2) idem  'sustained' is not apparent; 'aging' should preferably be replaced by 'skeletally mature'

Introduction

no comments

Results

3) were H2 levels in the culture system itself (not the incubator) checked? If yes, please report the H2 in experimental and control conditions for the in vitro cultures

4) secretory molecule profiles are known to be meaningless at their individual basis; did the authors determine TNF α and TGF β cytokine levels from macrophage cultures and determine their ratio to understand a shift from M1 to M2 subtype? Did the authors also include proper controls (for M1 and M2 macrophages) and check effects of H2 on their reversal?

5) TRAP staining for effects of H2 and conditioned medium of BMSC cultures requires quantification  why did the authors not quantify the supernatant of osteoclast cultures for TRAP enzyme activity?

6) H2 release from the nanoparticles requires cumulative quantification (in figure 2)

7) Authors are requested to demarcate the defect area in Figure 3b

8) Figures 3 and 5 require inclusion of overview histology (H&E stain)

Discussion

no additional comments

General

9) the authors are suggested to consult a native speaker to eliminate any textual/grammar/spelling errors from text as well as figures; the number of mistakes are not acceptable

Response to reviewers' comments

Reviewer #1 (Remarks to the Author):

The authors found that sustainable H₂ supply has the potential to remodeling the senescence microenvironment through unique and superior technology. Because H₂ exerts various effects in most organs, we predicted a role in the slowdown of the aging process. The authors clearly demonstrated the H₂ effect on anti-senescence by providing a large number of convincing experimental results. The technology is original and wonderful. This work definitely excels in our field. There are few publications related to H₂ on the role of H₂ in aging. The results are not simply descriptive, but satisfying a wide range of readers. This study fully supports the conclusions and claims. No additional experiments are required. There are no flaws in the analysis, interpretation or conclusions of the data. There is no reason to forbid publication and no revision is necessary. This work exceeds the standards expected in our field. Enough details are provided for anyone to reproduce the work. I have no additional remarks.

Response: Thank you very much for your especially high evaluation and for your recognition of the originality and excellence of this work.

Reviewer #2 (Remarks to the Author):

The authors have addressed many of my previous concerns from the Nature Aging review. However, I am still concerned that their phenotyping of senescence in vivo is incomplete. They do not provide any in vivo evidence for definitive markers of senescence (eg, telomere-associated foci or senescence-associated distension of satellite DNA). In most of the paper, they monitor senescence simply by expression of p16 - and this monitoring by flow appears to be still by a very questionable and unvalidated Santa Cruz p16 antibody (see below).

Response: Thank you very much for your kind reminder. According to your suggestion, we have carefully revised our manuscript again, and all the revised texts have been highlighted in green in the revised manuscript. We hope that our present revision can address your concerns.

Following your reminder, we have detected the telomere-associated foci as a definitive marker of senescence in vitro (Figure 1b) and in vivo (Figure S29). The related results further verify the anti-senescence effect of H₂, making the conclusion more convincing as you and we expected. The related experimental details, results and discussion have added in the revised manuscript (Line 1 in Page 5, Line 3–4 in Page 6, Line 9–11 in Page 11) and revised Supporting Information (Page 25).

It would help if they could provide some more direct link, even if in vitro, between H₂ and senescence. Specifically, do they have any evidence that H₂ kills senescent but not non-senescent cells (ie, is senolytic). Or is H₂ simply anti-inflammatory – and thus is functioning more as a “senomorphic” agent and reducing the SASP of senescent cells without killing them. And are the effects of H₂ specific (or relatively specific) for senescent cells, or is this just a general anti-inflammatory action of H₂. I believe the link to senescence is the weakest part of this paper and should be addressed in more detail.

Response: Thank you very much for your insightful comments. According to your reminder, we have checked the role of H₂ on the senescent osteocytes from 24-month-old mice. After incubation in the hydrogen incubator for 24 h, the cells were tested using the SA-β-gal kit and the CCK-8 kit. Meanwhile, senolytic (5 μM quercetin plus 1 μM dasatinib,) and senomorphic (1 μM ruxolitinib) agents are used as two controls. From Figure S9, both H₂ and senomorphic groups can significantly reduce SA-β-gal expression, suggesting a significant senomorphic effect. Surprisingly, we find that H₂ treatment can also slightly kill senescent cells, which is similar to the tumor-killing effect of H₂. In a word, it is discovered in this work that H₂ has both senomorphic and senolytic effects and plays a primary role of senomorphic agent. We have added the related data in the revised Supporting

Information (Figure S9 in Page 9), experimental details (Page 28) and discussion in the revised manuscript (Line 18–21 in Page 6).

Another significant issue is that they appear to use 3 different antibodies for p16 (Abcam on page 22, still a Santa Cruz – not validated – for flow (page 27) and Cell Signaling on page 28). This does make the critical flow data less credible and remains a significant concern.

Response: Many thanks for your kind reminder. We have repeated all the flow measurements by replacing p16 antibodies from Santa Cruz with ones from Abcam according to your suggestion. By comparison of repeated data, we find that there are indeed a little difference, but it has not changed the main conclusion of this work. Nevertheless, we would like to appreciate your kind reminder about antibody manufacturers. We have replaced the related data in the revised manuscript (Figure 3d–i in Page 10, Figure 5e–h in Page 15,) and Supporting Information (Figures S24–S28).

Reviewer #4 (Remarks to the Author):

The manuscript by Chen et al. describes the efficacy of H₂ (released from a biomaterial) to improve the SME, and hence regenerative capacity. The work seems interesting, but some issues apply:

Response: Thank you very much for your positive comment. According to your following suggestions, we have carefully revised our manuscript, and all the revised texts have been highlighted in blue in the revised manuscript.

Title

1) the words 'sustained' and 'aging' are preferable to be omitted from the title given the lack thereof in the described work

Response: Thank you for your kind suggestion. We agree with you, and therefore have omitted these two adjectives in the title of revised manuscript (Page 1).

Abstract

2) idem  'sustained' is not apparent; 'aging' should preferably be replaced by 'skeletally mature'

Response: Thanks a lot for your kind suggestion. We have omitted 'sustained' and replaced 'aging' with 'skeletally mature' in the abstract (Page 2).

Introduction

no comments

Results

3) were H₂ levels in the culture system itself (not the incubator) checked? If yes, please report the H₂ in experimental and control conditions for the in vitro cultures

Response: Thanks for your kind reminder. We indeed had measured the concentration of hydrogen in the cell culture medium to be nearly 0 μM in the general (without H₂ supplement) incubator, which was used as a control. We have added the related experimental results in Figure S34 in the revised Supporting Information (Page 27).

4) secretory molecule profiles are known to meaningless at their individual basis; did the authors determine TNF α and TGF β cytokine levels from macrophage cultures and determine their ratio to understand a shift from M1 to M2 subtype? Did the authors also include proper controls (for M1 and M2 macrophages) and check effects of H₂ on their reversal?

Response: Many thanks for your insightful suggestion. According to your reminder, we have executed the measurement of TNF- α and TGF- β 1 cytokine levels to determine their ratio for understanding a shift from M1 to M2 subtype. From Figure S8, the ratio of TNF- α level to TGF- β 1

level sharply declined to 0.17 from 4.14 after H₂ treatment for 7 days, indicating a H₂-induced great shift from M1 to M2 subtype of macrophages. We have added the related experimental details (Page 29) and results in the revised Supporting Information (Figure S8 in Page 8) and discussion in the revised manuscript (Line 16–18 in Page 6).

5) TRAP staining for effects of H₂ and conditioned medium of BMSC cultures requires quantification  why did the authors not quantify the supernatant of osteoclast cultures for TRAP enzyme activity?

Response: Thank you very much for your careful checking and kind reminder. According to your reminder, we have added the corresponding quantification data and quantified the supernatant of osteoclast cultures for TRAP enzyme activity. After 3-day treatment, the H₂ group can significantly decrease the activity of TRAP enzyme in osteoclast culture. The additional results have not changed the conclusion of this work, but indeed make it more convincing. We have added the related experimental results in the revised Supporting Information (Figure S13) and discussion in the revised manuscript (Line 1–2 in Page 7).

6) H₂ release from the nanoparticles requires cumulative quantification (in figure 2)

Response: Thanks a lot for your valuable suggestion. Following your suggestion, we have provided a cumulative quantification of H₂ release. The related results have been added in the revised Supporting Information (Figure S17).

7) Authors are requested to demarcate the defect area in Figure 3b

Response: We have demarcated the defect areas in Figure 3b (Page 10) and Figure 5ai (Page 15) according to your kind reminder.

8) Figures 3 and 5 require inclusion of overview histology (H&E stain)

Response: Unfortunately, the corresponding tissue sections which had been stained with multiple fluorescent agents cannot be stained again with HE. So we cannot provide the HE pictures at the corresponding sites in Figure 3b and Figure 5ai. But we are sure that the defect areas are in these fluorescence images as we demarcated in Figure 3b and Figure 5ai. Nevertheless, we would like to greatly appreciate your kind suggestion.

Discussion

no additional comments

General

9) the authors are suggested to consult a native speaker to eliminate any textual/grammar/spelling errors from text as well as figures; the number of mistakes are not acceptable

Response: We have double-checked and revised the manuscript.

REVIEWERS' COMMENTS

Reviewer #2 (Remarks to the Author):

The authors have satisfactorily addressed my concerns by performing additional studies and further modifying the paper. Thank you.

Reviewer #3 (Remarks to the Author):

The authors have addressed all but one of my issues adequately. The remaining issue is that histology solely based on high magnification sections (with an arbitrary line set to indicate defect margins) is inappropriate. As the authors clearly state in their m&m that 'Histology and immunofluorescence analysis - For histological analysis, retrieved femora were fixed in a 4% neutral paraformaldehyde buffer for 72 h, dehydrated, embedded, and sectioned at a 4.5 μm thickness. Sections were stained with hematoxylin and eosin (H&E) to examine their histological morphology.' providing overview histology of H&E sections of the bone defect is simple, but also extremely insightful.

Response to reviewers' comments

Reviewer #3 (Remarks to the Author):

The authors have addressed all but one of my issues adequately. The remaining issue is that histology solely based on high magnification sections (with an arbitrary line set to indicate defect margins) is inappropriate. As the authors clearly state in their m&m that 'Histology and immunofluorescence analysis - For histological analysis, retrieved femora were fixed in a 4% neutral paraformaldehyde buffer for 72 h, dehydrated, embedded, and sectioned at a 4.5 μm thickness. Sections were stained with hematoxylin and eosin (H&E) to examine their histological morphology.' providing overview histology of H&E sections of the bone defect is simple, but also extremely insightful.

Response: Thanks for your suggestion. We agree with the reviewer on this point. According to your suggestion, we have added the overview histology of H&E sections and related description and discussion in the revised Supporting Information (Figure S24 in Page 20) and in the revised manuscript (Line 5 in Page 8).